# Functional rescue of a disease-linked ERAD pathway mutation via alternative splicing

Huilun Helen Wang [1✉], Zhihong Wang [1], Liangguang Leo Lin [1], Sunil K Verma[1], Weronika Gniadzik[2], Hui Wang[1], Zexin Jason Li [1], Emily Whitestone[1], Lulu Jiang [2], Muge N Kuyumcu-Martinez [1,3,4,5], Shengyi Sun [6✉] & Ling Qi [1✉]

## Abstract

ER-associated degradation (ERAD) targets misfolded proteins in the endoplasmic reticulum (ER) for proteasomal degradation. Mutations in its most conserved branch involving the SEL1L-HRD1 complex cause ERAD-associated neurodevelopmental disorders with onset in infancy (ENDI), characterized by developmental delay, microcephaly, and locomotor dysfunction. Its most severe form, ENDI with agamma-globulinemia (ENDI-A), results from a bi-allelic SEL1L-Cys141Tyr (C141Y) mutation within its fibronectin II (FNII) domain and currently lacks effective treatment. Here, we find that knock-in mouse models carrying the C141Y mutation are unexpectedly rescued via increased use of an alternative splice donor within exon 4 leading to bypass of the mutant FNII-encoding region. The resulting SEL1L variant restores ERAD activity, and rescues perinatal lethality, B cell deficiency, and neurodevelopmental defects. Leveraging this mechanism, we demonstrate that antisense oligonucleotide-mediated exon skipping in patient-derived fibroblasts generates a truncated yet functional SEL1L protein that fully restores ERAD function and ER proteostasis. These results establish RNA splicing-modulation as a viable therapeutic strategy for ERAD deficiency and broaden the clinical potential of exon-skipping therapy to diseases of protein misfolding.

**Keywords** SEL1L-HRD1 ERAD; Disease Variant; ENDI-A; Alternative Splicing; Anti-sense Oligonucleotide
**Subject Categories** Molecular Biology of Disease; RNA Biology; Translation & Protein Quality

## Introduction

Nascent proteins in the endoplasmic reticulum (ER) that fail to fold properly are recognized as misfolded and subsequently targeted for cytosolic proteasomal degradation by an ER quality control system known as ER-associated degradation (ERAD) (Hampton et al, 1996; Sun and Brodsky, 2019; Wang et al, 2025; Ward et al, 1995). In mammals, the SEL1L-HRD1 protein complex represents the most conserved branch of ERAD (Bordallo et al, 1998; Christianson et al, 2023; Hampton et al, 1996; Wang et al, 2025). Misfolded proteins are first recognized by lectins such as OS9 and ERLEC1, and then delivered to SEL1L, which are then subsequently retrotranslocated through the HRD1 retrotranslocon followed by HRD1-mediated ubiquitination and proteasomal degradation (Christianson et al, 2008; Hosokawa et al, 2008; van der Goot et al, 2018; Wang et al, 2025). SEL1L is required for substrate recruitment (Cormier et al, 2009), HRD1 protein stability (Huang et al, 2013; Sun et al, 2014), and ERAD complex formation (Gardner et al, 2000; Lin et al, 2024; Mueller et al, 2008). Using various *Sel1l-* or *Hrd1-* knockout mouse models, we and others have shown that the SEL1L-HRD1 ERAD pathway plays vital roles in a range of physiological processes in a substrate- and cell-type-specific manner (Abdon et al, 2023; Bhattacharya and Qi, 2019; Bhattacharya et al, 2018; Hwang and Qi, 2018; Ji et al, 2016; Ji et al, 2023; Qi et al, 2017; Sha et al, 2014; Shrestha et al, 2020; Song et al, 2024; Sun et al, 2016; Thepsuwan et al, 2023; Wang et al, 2025; Wu et al, 2025; Xu et al, 2020; Yang et al, 2018; Yoshida et al, 2021).

The recent identification of human patients carrying bi-allelic *SEL1L* or *HRD1* variants has provided direct evidence of its critical role in humans. To date, eleven patients have been found carrying four distinct bi-allelic hypomorphic variants in *SEL1L* or *HRD1*, all presenting with moderate to severe intellectual disability, microcephaly, developmental delay, difficulty of feeding and locomotor dysfunction, collectively termed ERAD-associated neurodevelopmental disorders with onset in infancy (ENDI) (Wang et al, 2024; Weis et al, 2024). Among affected individuals, those carrying the SEL1L C141Y mutation (NM_005065.6: exon 4: c.422 G > A, p.Cys141Tyr) exhibit not only the core symptoms of ENDI but also agammaglobulinemia and early mortality. This more severe phenotype, termed ENDI-agammaglobulinemia (ENDI-A), has been associated with heightened ERAD dysfunction (Weis et al, 2024). However, despite these insights, direct evidence supporting the pathogenicity of this variant remains limited, and no therapeutic strategies have been established to date.

[1]Department of Molecular Physiology and Biological Physics, University of Virginia School of Medicine, Charlottesville, VA, USA. [2]Department of Neuroscience, Center for Brain Immunology and Glia (BIG), University of Virginia School of Medicine, Charlottesville, VA, USA. [3]Robert M. Berne Cardiovascular Research Center, University of Virginia School of Medicine, Charlottesville, VA, USA. [4]Center for RNA Science and Medicine, University of Virginia School of Medicine, Charlottesville, VA, USA. [5]Comprehensive Cancer Center, University of Virginia School of Medicine, Charlottesville, VA, USA. [6]Department of Pharmacology, University of Virginia School of Medicine, Charlottesville, VA, USA. ✉E-mail: kma8py@virginia.edu; bjk5fz@virginia.edu; xvr2hm@virginia.edu

In this study, while establishing the disease causality of the SEL1L C141Y mutation, we unexpectedly discovered that modulating splicing at exon 4 could reverse disease pathology in a knock-in (KI) mouse model carrying the mutation. Building on this finding, we successfully restored ERAD function in SEL1L C141Y patient-derived human fibroblasts using splice-switching antisense oligonucleotide (ASO)-mediated exon skipping. While modulating alternative splicing using ASOs—short, synthetic nucleotides that bind target RNAs to alter splicing or promote degradation—has emerged as a powerful therapeutic approach for genetic diseases (Crooke et al, 2021; Lauffer et al, 2024; Zhang et al, 2021), our findings extend this strategy to target ERAD dysfunction, an area previously unexplored.

# Results

## Alternative splicing of *Sel1L* exon 4 rescues lethality of the C141Y KI mice

C141 is located within the fibronectin II (FNII) domain, encoded by exon 4, of SEL1L, a ~50-amino acid motif of unknown function that is notably absent in yeast SEL1L ortholog Hrd3 (Harada et al, 1999; Weis et al, 2024) (Fig. 1A,B). This domain is stabilized by two disulfide bonds (Cys127-Cys153 and Cys141-Cys168) (Pickford et al, 2001) (Fig. 1B). To investigate the disease causality and pathological consequences of *SEL1L* C141Y mutation in vivo, we generated KI mouse models using CRISPR-Cas9 genome editing (Fig. EV1A). Sanger sequencing confirmed successful introduction of the intended G-to-A mutation in the genomes of three independent founder lines (Lines A–C), which were generated from separate CRISPR-Cas9 embryo injections (Figs. 1C and EV1B). From these lines, we subsequently obtained WT, heterozygous (HET), and homozygous KI offspring (Fig. EV1C,D). Unexpectedly, only Line C produced viable homozygous mice (hereafter KI') at the expected ratio that survived past postnatal day 21 (p21) at Mendelian ratios (Fig. 1D). In contrast, Lines A and B yielded only 5 viable homozygous mice out of ~280 pups surviving past p21 with an overall survival rate of ~1.8% (Fig. 1D). Given the shared phenotypes between Lines A and B, they were grouped as the KI line for subsequent analyses using tissues from p0 neonates due to lethality. HET mice from all lines were viable (Fig. 1D) and appeared phenotypically indistinguishable from WT littermates.

Interestingly, using the F1R1 PCR primer set spanning all 21 exons of SEL1L, we found that mouse *Sel1L* naturally expresses two mRNA isoforms in the liver (Fig. 1E,F, lanes 2 and 4). Sequencing revealed that these isoforms resulted from alternative splicing of exon 4: a predominant full-length isoform (*Sel1L-a*, ~80%), and a shorter isoform (*Sel1L-b*) lacking 150 bp due to the use of an alternative "GU" splice donor site within exon 4 (arrow, Fig. 1G; illustrated in Fig. 1E). This splicing event removed amino acids 118–166, eliminating most of the FNII domain, including the C141Y mutation site (Fig. 1E). Notably, using the F2R2 primer set flanking exon 4 (Fig. 1E), we found that despite large variations in overall *Sel1L* expression levels across WT mouse tissues (for example, a fivefold difference between pancreas and spleen or lungs), the relative abundance of isoforms *a* and *b* remained consistent, with isoform *b* comprising ~20% of total *Sel1L* transcript (Figs. 1H and EV2A). Moreover, this splice pattern in

the liver was unaffected by ER stress induced by tunicamycin injection (Fig. EV2B).

Sequence analysis confirmed that no off-target mutations were present in the *Sel1L* cDNA aside from the engineered mutations (Appendix Fig. S1). Remarkably, KI mice predominantly expressed the full-length *Sel1L-a* isoform, while the KI' line expressed exclusively the shorter *Sel1L-b* isoform (Fig. 1F). In the KI' line, the alternative splicing occurred at the same internal alternative "GU" splice donor site within exon 4, thereby bypassing the pathogenic C141Y mutation (Fig. 1G). To further confirm this splicing pattern, we performed RT-PCR using the alternative primer pair flanking exon 4 (F2R2, Fig. 1E) and resolved the PCR products by acrylamide gel electrophoresis for enhanced small-fragment resolution (Fig. 1I). While WT livers showed a 4:1 ratio of *Sel1L-a* to *-b*, KI and KI' livers almost exclusively expressed *Sel1L-a* and *-b*, respectively (Fig. 1I,J). Using an additional primer pair F3R3 spanning exons 1 and 6 (Fig. EV2C), we further confirmed the splicing patterns observed above (Fig. EV2C–E). Notably, HET mice from both lines displayed intermediate isoform expression levels, falling between those of WT and the respective KI lines (Fig. EV2D,E). Comparable *Sel1L* splicing patterns were detected across multiple tissues in WT, KI, and KI' mice (Fig. 1K,L).

## Alternative splicing of *Sel1L* exon 4 rescues ERAD defects in KI' mice

To assess the consequences of exon 4 skipping at the protein level, we measured SEL1L expression using two antibodies targeting distinct epitopes (Weis et al, 2024) (Fig. 2A). KI livers exhibited a near-complete loss of the full-length SEL1L-A expression normally present in WT tissues. In contrast, KI' livers expressed the shorter SEL1L-B isoform at ~50% of WT SEL1L-A levels (Fig. 2B). SEL1L-B is ~5 kDa (i.e., 50 aa) smaller than the full-length isoform A, consistent with skipping of 150 bp exon 4. Consistent with ERAD deficiency, KI mice exhibited a ~50% reduction in the E3 ligase HRD1, ~80% decrease in the ER lectin ERLEC1, and a ~fivefold accumulation of the known ERAD substrate and UPR sensor IRE1α (Hetz et al, 2020; Hwang and Qi, 2018; Sun et al, 2015) (Fig. 2C). Remarkably, KI' livers, despite expressing only ~50% of WT SEL1L-A levels, exhibited normal HRD1 and ERLEC1 levels and prevented IRE1α accumulation (Fig. 2C), indicating preserved ERAD function in these mice. Quantification is shown in Fig. 2D,E. As expected for a recessive trait, HET mice from both lines expressed intermediate SEL1L isoform levels relative to WT and their respective KI alleles (Fig. 2D,F), consistent with a Mendelian recessive inheritance model (Weis et al, 2024).

We next examined whether SEL1L-B remains functionally engaged in ERAD. Immunoprecipitation experiments showed that endogenous SEL1L-B interacted with major ERAD components, including HRD1 and OS9, with efficiency comparable to the full-length SEL1L-A isoform (Fig. 2G). Endoglycosidase H (Endo H) and Peptide:N-glycosidase F (PNGase F) treatment further demonstrated that, like SEL1L-A, SEL1L-B was glycosylated and fully Endo H-sensitive, indicating that it is retained within the ER (Fig. 2H). Taken together, these findings show that alternative splicing of *Sel1L* exon 4 produces the SEL1L-B isoform, which retains ER localization and ERAD function.

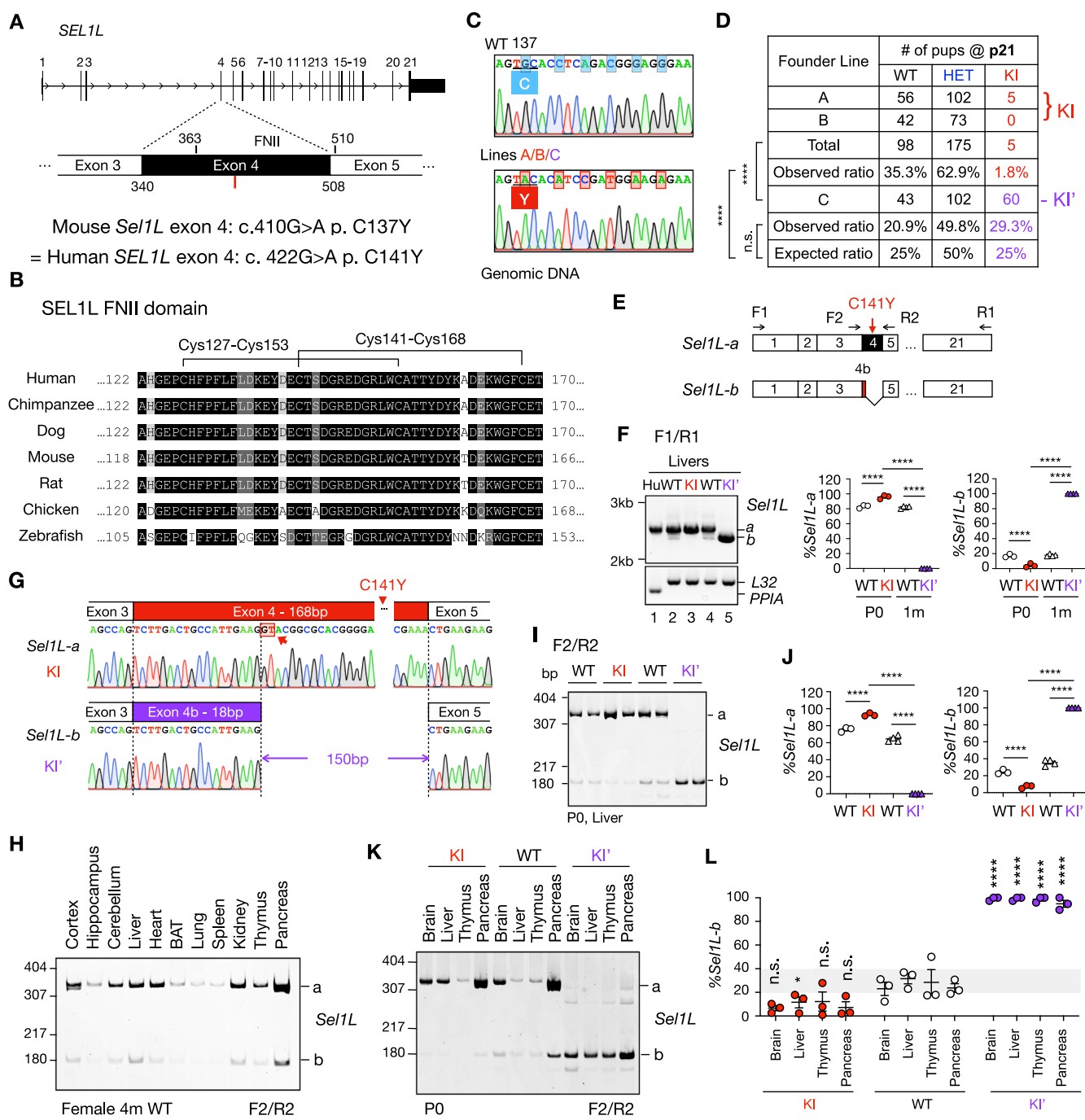

## Disruption of the intronic splice donor enables alternative splicing in KI′ mice

To investigate the mechanism underlying exon 4 skipping, we sequenced the genomic region surrounding *Sel1L* exon 4. This analysis identified a single-nucleotide (T) deletion at the canonical "GT" splice donor site at the exon 4-intron 4 junction (Fig. 3A), likely introduced during the CRISPR/Cas9-mediated gene editing. To test whether this disruption alone is sufficient to promote alternative splicing (Fig. 3B), we constructed a splicing minigene

reporter (Belanger et al, 2018) containing 168 bp *Sel1L* exon 4 flanked by segments of 378 bp introns 3 and 463 bp intron 4, inserted between two GFP fragments (Fig. 3C). RT-PCR using GFP-specific primers (F4/R4, Fig. 3D) revealed that the WT minigene produced both full-length exon 4 and truncated exon 4b isoforms, mirroring the in vivo pattern (Fig. 3E,F). Mutation of the internal alternative splice donor site (mutation #1, GT → GC) abolished the exon 4b isoform, confirming its functionality (Fig. 3E,F). However, deletion of the canonical splice donor site (mutation #2, GT → G-) resulted exclusively in complete exon

**Figure 1.  Alternative splicing of Sel1L exon 4 rescues the lethality of SEL1L C141Y knockin (KI) mice.**

(A) A diagram showing the location of *SEL1L* C141Y mutation (red line) and Fibronectin II (FNII) domain in the *SEL1L* gene. (B) Sequence alignment of the FNII domain across vertebrate species. The darkness of the shading indicates the degree of conservation, with black representing the highest conservation of the residue side chain property. Lines indicate conserved disulfide bond pairs. (C) Sanger sequencing of *Sel1L* genes from the homozygous progenies of WT and KI founder lines A/B/C. Red shaded box highlights the mutation identified in exon 4. (D) The survival rate of KI/KI' pups and their WT and HET littermates at postnatal day 21 (p21). (E) The diagram of the *Sel1L* isoforms and the primer design. (F) DNA agarose gel electrophoresis analysis of full-length SEL1L transcript from a healthy human liver biopsy sample and *Sel1L* transcripts from WT, KI and KI' mouse livers at ages of postnatal day 0 (p0) and 1 month (1 m), respectively, with quantification shown on the right. *PPIA* and *L32*, internal controls for human and mouse tissues, respectively. $n = 3$–4 mice/group. %*Sel1L-a*: WT vs. KI, ****$P < 0.0001$; WT vs. KI', ****$P < 0.0001$; KI vs KI', ****$P < 0.0001$. %*Sel1L-b*: WT vs. KI, ****$P < 0.0001$; WT vs. KI', ****$P < 0.0001$; KI vs KI', ****$P < 0.0001$. (G) Sanger sequencing of *Sel1L* cDNA, focusing on exon 4 and splicing junctions. Red arrow and shaded red box, the alternative GT donor. (H) DNA polyacrylamide gel electrophoresis (PAGE) analysis of *Sel1L* exon 4 splicing in various mouse tissues, with quantification shown in Fig. EV2A. BAT, Brown adipose tissue. (I, J) DNA PAGE analysis (I) of *Sel1L* exon 4 isoforms in mouse livers at indicated ages with quantification shown in (J). $n = 3$–4 mice/group. %*Sel1L-a*: WT vs. KI, ****$P < 0.0001$; WT vs. KI', ****$P < 0.0001$; KI vs KI', ****$P < 0.0001$. %*Sel1L-b*: WT vs. KI, ****$P < 0.0001$; WT vs. KI', ****$P < 0.0001$; KI vs KI', ****$P < 0.0001$. (K, L) DNA PAGE analysis (K) of *Sel1L* exon 4 splicing in the brain, liver, thymus and pancreas from WT, KI and KI' p0 pups, with quantification shown in (L). Statistics indicates the comparison between the same tissue type of WT vs. KI or KI'. $n = 3$ mice/group. WT vs. KI: Brain, $P = 0.056$; Liver, *$P = 0.019$; Thymus, $P = 0.059$; Pancreas, $P = 0.055$. WT vs. KI': Brain, ****$P < 0.0001$; Liver, ****$P < 0.0001$; Thymus, ****$P < 0.0001$; Pancreas, ****$P < 0.0001$. Data are represented as means ± SEM. n.s. not significant. *$P < 0.05$, ****$P < 0.0001$, by Chi-square test for (D); two-way ANOVA followed by Tukey's multiple-comparisons test for (F, J), one-way ANOVA followed by Tukey's post hoc test for (L). Source data are available online for this figure.

4 skipping (ΔExon 4), rather than preferential use of the internal alternative donor, unlike what is observed in vivo (Fig. 3E,F). Quantification is shown in Fig. 3G.

This discrepancy prompted us to examine whether additional sequence changes introduced during CRISPR/Cas9 editing, specifically the C141Y mutation and nearby 5 synonymous mutations used for genotyping (Fig. 3H), influence splice site selection. We introduced either the C141Y mutation (mutation #3, G → A) or the synonymous nucleotide changes (mutation #4, C..A..C..G..G → A..C..T..A..A) into the WT minigene or into the minigene carrying mutation #2 (Fig. 3H). The C141Y mutation did not affect splicing. However, the synonymous mutations reduced usage of alternative splice donor when the canonical site was intact, but enhanced usage of the alternative splice donor when the canonical site was disrupted (Fig. 3I,J). These results align with our in vivo findings, in which the KI and KI' mice predominantly use the canonical splice donor and the internal alternative donor, respectively. Together, these data suggest that loss of the canonical splice donor site creates a permissive environment for exon skipping, and that the synonymous mutations introduced during gene editing further bias splice site choice toward the alternative donor within exon 4.

## Disease casuality of SEL1L C141Y and pathophysiological significance of *Sel1L* alternative splicing

To assess the physiological relevance of *Sel1L* alternative splicing, we phenotypically characterized and compared KI and KI' mice. While very few KI pups could survive past weaning (Fig. 1D), they were born, albeit at a reduced frequency of 16% at p0, indicating that the mutation did not cause complete embryonic lethal (Fig. 4A). At birth, KI pups were indistinguishable from WT littermates in body weight, gross morphology, and tissue mass (Fig. 4B–D). However, by 12 h postpartum, KI neonates lacked visible milk sacs (arrows in Fig. 4C,E). These phenotype parallels clinical reports of severe feeding difficulties in ENDI-A patients shortly after birth (Weis et al, 2024). Within the defined cohort, all KI pups died within 48 h of birth (Fig. 4F), although rare survivors were observed in separate litters (5 out of 290 pups, Fig. 1D). In contrast, KI' mice were born at expected Mendelian ratios and showed normal growth comparable to WT and HET littermates (Fig. 4G).

ENDI-A patients exhibit profound B cell lymphopenia (Weis et al, 2024), akin to mice with developing B cell-specific deletion of *Sel1L* or *Hrd1* (Ji et al, 2016; Yang et al, 2018). Similarly, KI neonates exhibited marked reductions in mature B cells in peripheral blood and spleen at p0, as measured by flow cytometry, and immunofluorescence staining using mature B cell markers CD19 and B220 (Figs. 4H–J and EV3A,B). By contrast, KI' mice showed normal circulating B cell numbers at birth (Figs. 4H and EV3C) and puberty (Fig. EV3D,E).

Given that ENDI patients also exhibit intellectual disability (Wang et al, 2024; Weis et al, 2024), we next examined cortical development at p0. Similar to our observations in the liver (Fig. 2B), KI brains exhibited a near-complete loss of SEL1L expression, an ~70% reduction of HRD1 protein, and an ~eight-fold increase in IRE1α levels (Fig. 5A–C). All of these abnormalities were fully restored in KI' mice (Fig. 5D,E). Notably, despite its marked accumulation, IRE1α was not overtly activated in KI brains, as evidenced by the absence of IRE1α phosphorylation on the Phos-tag gel (Qi et al, 2011; Yang et al, 2010) and the lack of enhanced *XBP1* mRNA splicing, the downstream output of IRE1α signaling (Figs. 5F,G and EV4A). Consistent with this, we observed no significant induction of ER chaperones (BiP and PDI) and no phosphorylation of another UPR sensor, PERK (Fig. EV4B,C). Together, these data indicate minimal, if any, UPR activation in KI brains.

Cortical development occurs during embryogenesis and results in the formation of six distinct layers arranged in an inside-out pattern from Layer VI to Layer I (Stiles and Jernigan, 2010). In WT brains, immunofluorescence staining revealed well-organized cortical layering, with clearly defined Layer II/III (SATB2-positive) and Layer V (CTIP2-positive) neurons (Fig. 5H,I). In contrast, KI mice exhibited disrupted cortical architecture, including ectopic localization of SATB2-positive cells below Layer V without obvious cell loss, indicative of impaired neuronal migration or cortical specification (Fig. 5H,I). Interestingly, KI' mice displayed a normalized SATB2+ cell distribution and restored laminar organization (Fig. 5H,I). Collectively, these findings establish pathogenic nature of the *SEL1L* C141Y variant and demonstrate that exon 4 skipping via alternative splicing functionally rescue the lethal and developmental defects associated with the mutation in vivo.

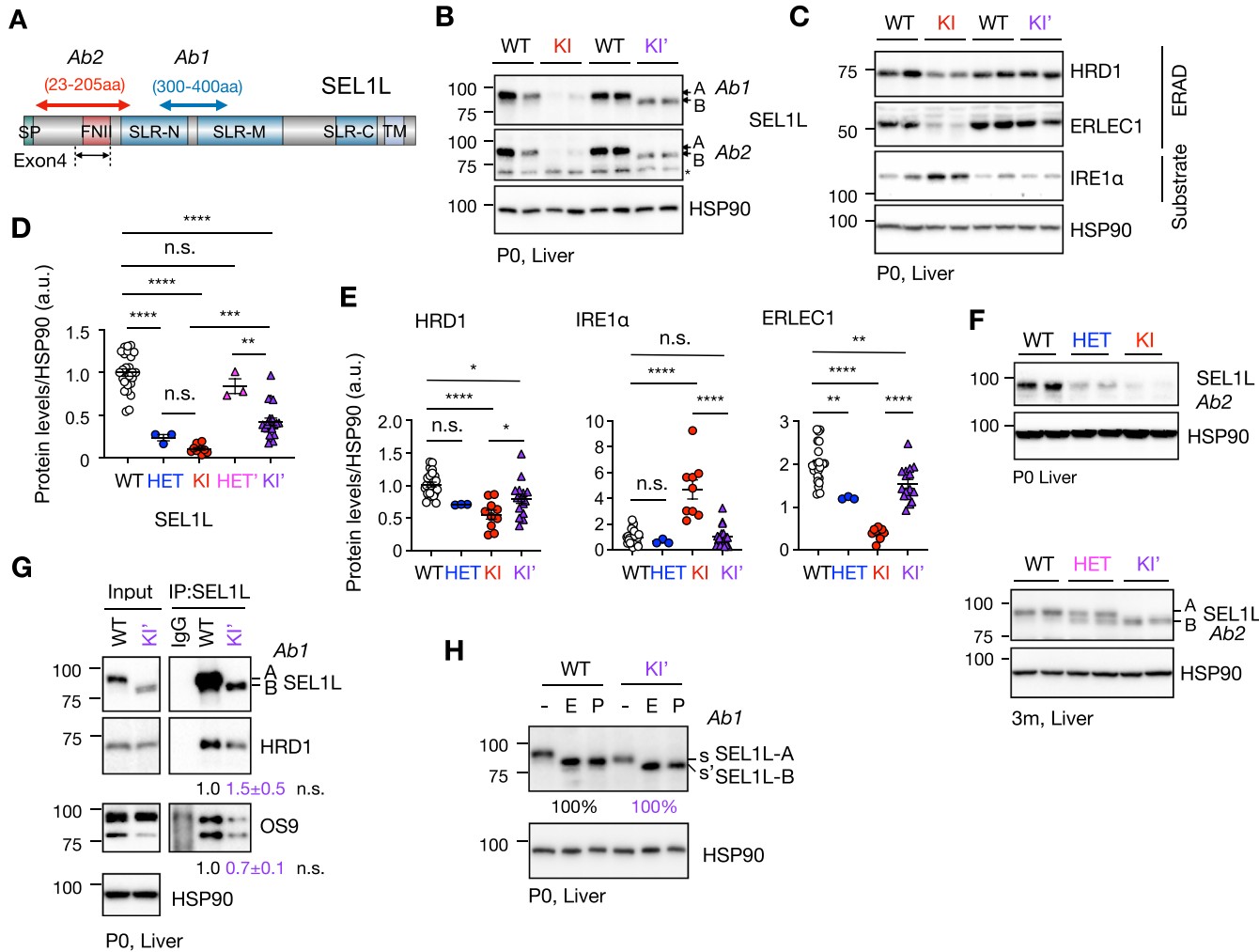

**Figure 2. Alternative splicing of Sel1L exon 4 rescues ERAD dysfunction caused by SEL1L C141Y homozygous mutation.**

(A) Diagram showing the epitope target regions of the home-made (Ab2) and Abcam (Ab1) anti-SEL1L antibodies on the SEL1L protein in (B, F). (B–F) Western blot analysis of SEL1L (B, F), ERAD proteins and known ERAD substrate IRE1α (C) in WT, HET, KI, HET' and KI' mouse livers with quantification in (D, SEL1L: WT vs. HET, ****$P < 0.0001$; WT vs. KI, ****$P < 0.0001$; WT vs. HET', $P = 0.62$; WT vs. KI', ****$P < 0.0001$; HET vs. KI, $P = 0.83$; KI vs. KI', ***$P = 0.0004$; HET' vs. KI', **$P = 0.0053$) and (E, HRD1: WT vs. HET, $P = 0.11$; WT vs. KI, ****$P < 0.0001$; WT vs. KI', *$P = 0.020$; KI vs. KI', *$P = 0.042$. IRE1α: WT vs. HET, $P = 0.96$; WT vs. KI, ****$P < 0.0001$; WT vs. KI', $P > 0.99$; KI vs. KI', ****$P < 0.0001$. ERLEC1: WT vs. HET, **$P = 0.0071$; WT vs. KI, ****$P < 0.0001$; WT vs. KI', **$P = 0.0026$; KI vs. KI',****$P < 0.0001$), respectively. $n = 4$–20 mice/group. SEL1L from both antibodies were quantitated as average in (D). Astericks, non-specific bands. (G) Immunoprecipitation of endogenous SEL1L from p0 WT and KI' livers, with quantification of interactions with ERAD components normalized to SEL1L levels. $n = 3$ biological replicates. (H) Representative western blot of SEL1L proteins from p0 WT or KI' liver lysates treated with or without EndoH (E) or PNGase (P) treatment, with quantification of the percentage of Endo H-sensitive band shown below the blot. $n = 3$ independent biological replicates. s and s', sensitive band for WT and KI' line, respectively. Data are represented as means ± SEM. n.s., not significant. *$P < 0.05$; **$P < 0.01$; ***$P < 0.001$, ****$P < 0.0001$, by one-way ANOVA followed by Tukey's post hoc test for (D, E), two-tailed $t$ test for (G). Source data are available online for this figure.

## ASO-mediated exon skipping rescues ERAD defects in patient-derived fibroblasts

Prompted by serendipitous observations in mice, we next explored whether an exon-skipping strategy could similarly be applied to human cells harboring the C141Y mutation. Interestingly, sequence alignment revealed that the internal "GU" splice donor site found in murine exon 4 is not conserved in humans or other species (Fig. 6A), consistent with the exclusive expression of the full-length *SEL1L-A* isoform in human tissues (lane 1, Fig. 1F). To test the feasibility of inducing exon skipping,

we treated fibroblasts from a SEL1L C141Y patient with a panel of 25-nucleotide-long ASOs targeting exon 4 and its flanking intronic regions (Fig. 6B). RT-PCR using F5R5 primer pair (Fig. 6C), followed by Sanger sequencing, revealed the appearance of two alternative splice variants in addition to the full-length isoform A: isoform *B*, induced by ASO1 and ASO3-5, which lacked the entire exon 4 (distinct from the mouse *Sel1L-b* isoform); and isoform *C* detected with ASO6-7, which lacked the first 63 bp of exon 4, likely due to disruption of the intron 3 splice acceptor and activation of a cryptic splice site within exon 4 (Fig. 6C,D).

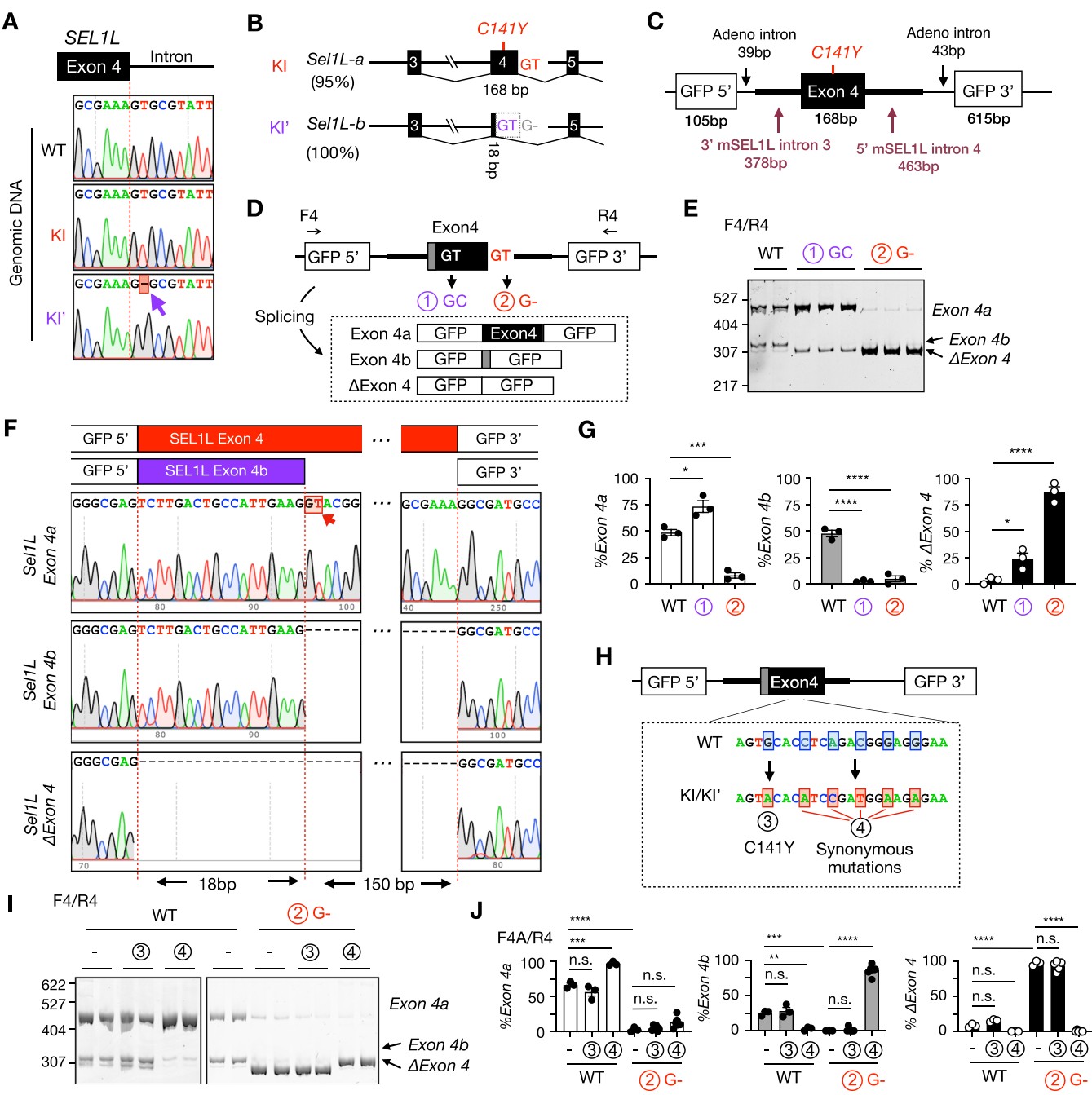

**Figure 3. The mutation of major splicing donor enables the usage of an internal alternative splice donor in mice.**

(A) Sanger sequencing results of *Sel1L* exon 4-intron 4 junction. Mutation was indicated by the red shaded box and purple arrow. (B) Schematic of the alternative splicing of *Sel1L* exon 4 with the percentage of isoforms in KI and KI' mice indicated on the left. (C) Diagram of the splicing minigene reporter. (D, H) Diagrams of the indicated mutations in the minigene reporter construct, including primer design and the expected splicing products. (E, G) DNA PAGE analysis (E) of the WT and mutated minigene reporters as indicated in (D) by using primer pair F4/R4, with quantification in (G). $n = 3$ independent biological replicates. *%Exon 4a*: WT vs. #1, $*P = 0.011$; WT vs. #2, $***P = 0.0008$. *%Exon 4b*: WT vs. #1, $****P < 0.0001$; WT vs. #2, $****P < 0.0001$. *%ΔExon 4*: WT vs. #1, $*P = 0.042$; WT vs. #2, $****P < 0.0001$. (F) Sanger sequencing analysis for the splice products shown in (E). (I, J) DNA PAGE analysis of the WT and mutated minigene reporters as indicated in (H) by using primer pair F4/R4, with quantification in (J). $n = 3–6$ independent biological replicates. *%Exon 4a*: WT(−) vs. WT(#3), $P = 0.37$; WT(−) vs. WT(#4), $***P = 0.0001$; WT(−) vs. #2(−), $****P < 0.0001$; #2(−) vs. #2(#3), $P > 0.99$; #2(−) vs. #2(#4), $P = 0.21$. *%Exon 4b*: WT(−) vs. WT(#3), $P > 0.99$; WT(−) vs. WT(#4), $**P = 0.0010$; WT(−) vs. #2(−), $***P = 0.0003$; #2(−) vs. #2(#3), $P = 0.99$; #2(−) vs. #2(#4), $****P < 0.0001$. *%ΔExon 4*: WT(−) vs. WT(#3), $P = 0.18$; WT(−) vs. WT(#4), $P = 0.12$; WT(−) vs. #2(−), $****P < 0.0001$; #2(−) vs. #2(#3), $P = 0.80$; #2(−) vs. #2(#4), $****P < 0.0001$. Data are represented as means ± SEM. n.s., not significant. $*P < 0.05$; $**P < 0.01$; $***P < 0.001$, $****P < 0.0001$, by one-way ANOVA followed by Tukey's post hoc test for (G), two-way ANOVA followed by Tukey's multiple-comparisons test for (J). Source data are available online for this figure.

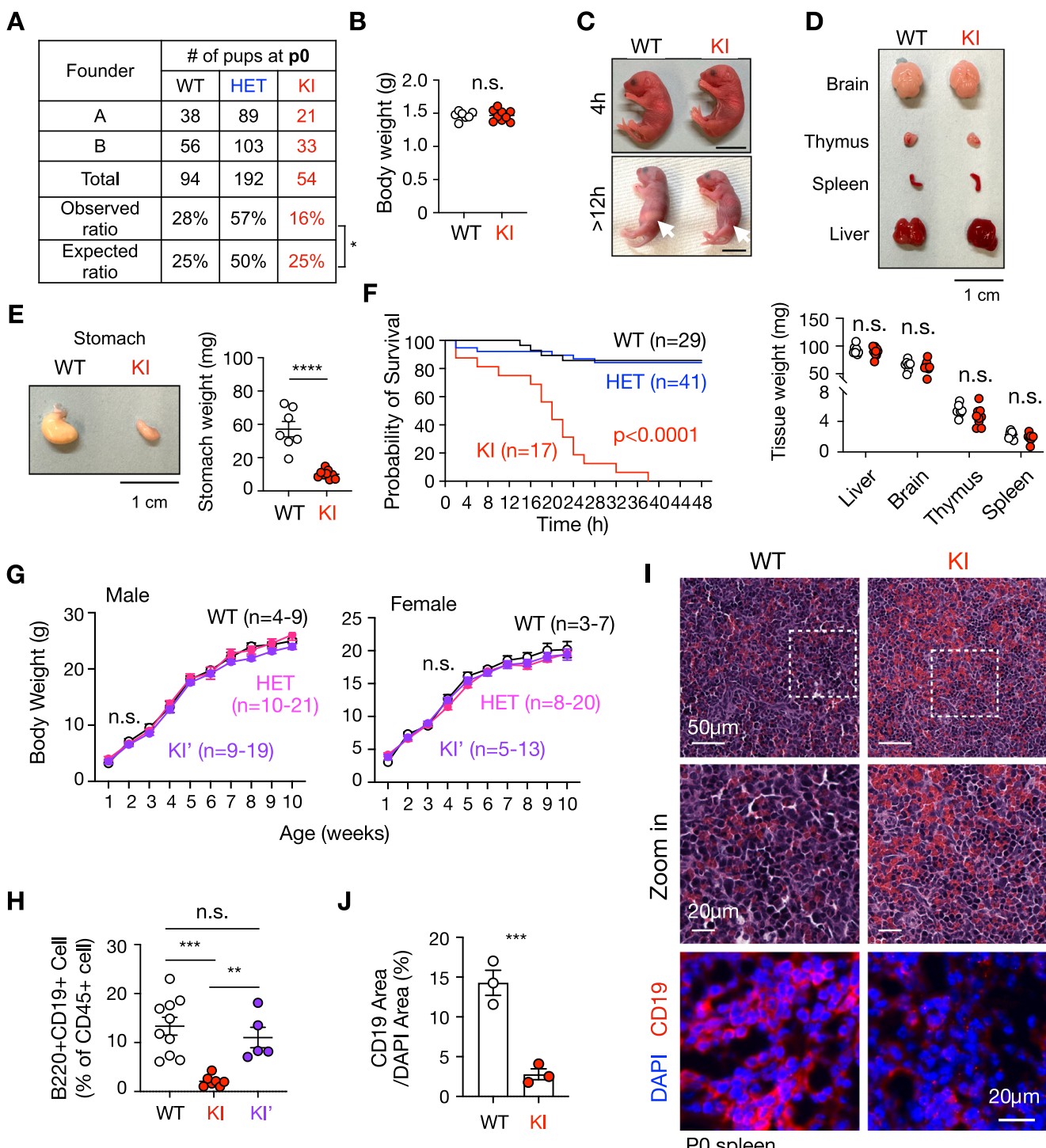

**A**

| Founder | # of pups at **p0** | | |
|---|---|---|---|
| | WT | HET | KI |
| A | 38 | 89 | 21 |
| B | 56 | 103 | 33 |
| Total | 94 | 192 | 54 |
| Observed ratio | 28% | 57% | 16% |
| Expected ratio | 25% | 50% | 25% |

**B** Body weight (g) — n.s. WT, KI

**C** WT KI — 4h, >12h

**D** WT KI — Brain, Thymus, Spleen, Liver — 1 cm

Tissue weight (mg) — n.s. Liver, Brain, Thymus, Spleen

**E** Stomach — WT KI — 1 cm — Stomach weight (mg) **** WT KI

**F** Probability of Survival — WT (n=29), HET (n=41), KI (n=17), p<0.0001 — Time (h)

**G** Male — Body Weight (g) — WT (n=4-9), HET (n=10-21), KI' (n=9-19) — n.s. — Age (weeks)

Female — WT (n=3-7), HET (n=8-20), KI' (n=5-13) — n.s. — Age (weeks)

**H** B220+CD19+ Cell (% of CD45+ cell) — n.s., ***, ** — WT, KI, KI'

**J** CD19 Area /DAPI Area (%) *** — WT, KI

**I** WT KI — Zoom in — 50µm, 20µm — DAPI CD19 — 20µm — P0 spleen

Among all the tested ASOs, ASO1 was the most effective, inducing exon 4 skipping in ~60% of transcripts in both WT and patient-derived fibroblasts (Figs. 6C,E and 7A). Neither the SEL1L C141Y mutation nor ASO treatment altered overall *SEL1L* mRNA levels (Fig. 7B). Comparable dose-dependent exon-skipping results were observed in ASO-treated WT HEK293T cells (Fig. 7C). ASO1 treatment significantly inceased expression of the truncated SEL1L-B protein in WT fibroblasts, accounting for ~40% of total SEL1L protein, and rescued SEL1L expression in C141Y patient fibroblasts (Fig. 7D,E). Consistent with the role of SEL1L in stabilizing HRD1

(Sun et al, 2014), ASO1 treatment restored HRD1 protein levels and rescued ERAD function, as evidenced by decreased accumulation of known ERAD substrates including IRE1α and CD147 (Tyler et al, 2012) (Fig. 7D,F), and reduced *XBP1* mRNA splicing (Fig. 7G). This is consistent with our previous data that complete deletion of the FNII domain produced a truncated SEL1L protein (~5 kDa smaller than WT) that preserved normal ERAD function (Weis et al, 2024). These results indicate that the domain itself is dispensable for ERAD function. Taken together, these findings demonstrate that ASO-mediated exon skipping restores functional

**Figure 4. The alternative splicing of Sel1L exon 4 rescues perinatal lethality and B cell lymphopenia in SEL1L C141Y KI mice.**

(A) The survival rate of WT, HET and KI pups at postnatal day 0 (p0). Observed vs. Expected *$P = 0.013$. (B) Body weight at p0. $n = 7$–9 mice/group. WT vs. KI, $P = 0.94$. (C) The photo of pups at postnatal 4 h (upper) and >12 h (lower). Scale bars: 1 cm. (D) Tissue morphology of SEL1L C141Y KI pups at P0, with tissue weight quantification shown below. $n = 7$–9 mice/group. Scale bars: 1 cm. WT vs. KI, Liver $P = 0.45$, Brain $P = 0.65$, Thymus $P = 0.14$, Spleen $P = 0.16$. (E) Morphology of postprandial stomach in the P0 WT and KI pup, with quantification shown on the right. $n = 7$–9 mice/group. Scale bars: 1 cm. WT vs. KI, ****$P = 0.000013$. (F) The survival curve of WT, HET and KI pups after birth. $n = 17$–41 mice/group. WT vs. HET, $P = 0.86$; WT vs. KI, ****$P < 0.0001$. (G) Body weight growth of male and female WT, HET, KI' mice for postnatal week 1 to week 10. $n = 3$–20 mice/group. (H) Quantification of B cells as a percentage of CD45$^+$ peripheral blood mononuclear cells (PBMCs) at p0 by flow cytometry in Fig. EV3B,C. $n = 5$–10 mice/group. WT vs. KI, ***$P = 0.0002$; WT vs. KI', $P = 0.63$; KI vs. KI', **$P = 0.0091$. (I, J) Hematoxylin and eosin (H&E) staining and Immunofluorescence of CD19$^+$ B cells in p0 mouse spleens from SEL1L C141Y KI pups at p0, with quantification shown in (J). Red, CD19; Blue, DAPI. $n = 3$ mice/group. Scale bars: 50 μm for zoom-out photo, 20 μm for zoom-in photo and immunofluorescence figure. ***$P = 0.0026$. Data are represented as means ± SEM. n.s. not significant. *$P < 0.05$; **$P < 0.01$; ***$P < 0.001$, ****$P < 0.0001$, by Chi-square test for (A); two-tailed $t$ test for (B, D, E, J); Mantel–Cox test for (F); two-way ANOVA followed by Tukey's multiple-comparisons test for (G); one-way ANOVA followed by Tukey's post hoc test for (H). Source data are available online for this figure.

SEL1L and rescues ERAD defects in patient-derived fibroblasts, highlighting its therapeutic potential for patients with SEL1L defects.

## Discussion

We recently identified bi-allelic mutations in *SEL1L* and *HRD1* in patients with ENDI and ENDI-A (Wang et al, 2024; Weis et al, 2024), implicating SEL1L-HRD1 ERAD in the pathogenesis of these rare but severe genetic disorders. Using a KI mouse model, we established the pathogenicity of the SEL1L C141Y variant located in the FNII domain. Moreover, we found that, although this residue is essential for SEL1L protein stability, the FNII domain itself is dispensable for SEL1L and ERAD function. This unexpected finding, rooted in the phenotypic discrepancies among three independently generated KI mouse lines carrying the same mutation, led to the discovery of naturally occurring alternative splicing as a mechanism of functional rescue (Fig. 8A,B). Thus, our study not only defines the molecular pathology of the SEL1L C141Y mutation but also highlights splicing modulation as a promising therapeutic strategy for correcting ERAD defects.

The SEL1L FNII domain is conserved across vertebrates and is also found in proteins such as fibronectin and tissue-type plasminogen activator/coagulation factor XII (Harada et al, 1999; Weis et al, 2024), but it is absent in yeast SEL1L counterpart Hrd3. This pattern suggests that the domain may have been evolutionarily acquired during vertebrate evolution through exon shuffling (Patel et al, 1987). In rodent SEL1L, this domain is encoded by exon 4, which undergoes alternative splicing to generate isoform diversity – a feature not observed in humans. This rodent-specific splicing event may provide an added layer of control over SEL1L function. Although the precise role of the FNII domain remains unclear, our data indicate that SEL1L-B, which lacks this domain, is less stable than the full-length SEL1L-A protein, suggesting a role in stabilizing SEL1L. In addition, a recent study implicated SEL1L in collagen turnover and proposed that its FNII domain may contribute to collagen binding (Podolsky et al, 2024). Further proteomic studies including ASO-treated patient samples or interactome analyses of different SEL1L isoforms will be needed to delineate its precise function and broader biological relevance.

Our minigene assay demonstrated that the synonymous mutations introduced in the KI mice participate in splicing regulation. Sequence analysis suggests that these synonymous changes may have inadvertently created an exonic splicing enhancer (ESE) motif recognized by Serine/Arginine-rich (SR)

proteins (Long and Caceres, 2009). In the wild-type sequence (*AGTGCACCTCAGACGGGAGGGAA*), the purine-rich region (*GGAGGGAA*) does not strongly resemble canonical ESE consensus motifs typically bound by SR proteins. In contrast, the synonymous mutations in the KI line (*AGTACACATCCGATG-GAAGAGAA*) introduces a core element (*GAAGAGAA*) that closely matches well-characterized ESE motifs such as *GAAGAA* or *GAAGAAG* (Tacke and Manley, 1995), which are recognized by SR proteins including SRSF1 and TRA2-β (Long and Caceres, 2009; Shepard and Hertel, 2009). This likely enhances exon recognition and promotes exon inclusion in the presence of the synonymous mutations (Ibrahim et al, 2005). Further investigation will be needed to confirm the functional ESE motif and to identify the specific SR proteins involved in this splicing regulation.

The clinical success of ASO therapies, such as eteplirsen for Duchenne muscular dystrophy and approved treatments for spinal muscular atrophy and amyotrophic lateral sclerosis, underscores their utility in correcting genetic defects by restoring partially functional proteins with high specificity and minimal off-target effects (Crooke et al, 2021; Kim et al, 2022; Lim et al, 2017). Intrathecal delivery of ASOs has shown promise in treating rare neurodevelopmental disorders, with substantial improvements in seizures and cognitive outcomes when administered early (Chen et al, 2024; Sztainberg et al, 2015; Wagner et al, 2025; Ziegler et al, 2024). To date, ASOs have not been applied to diseases involving ERAD dysfunction. Our findings establish a proof-of-concept for this approach, demonstrating that targeted exon skipping can restore proteostasis in the setting of a pathogenic *SEL1L* mutation. Given the essential role of SEL1L-HRD1 ERAD in cellular homeostasis, this strategy may hold promise not only for *SEL1L*-related disorders but also for a broader range of protein misfolding diseases.

## Methods

**Reagents and tools table**

| Reagent/resource | Reference or source | Identifier or catalog number |
|---|---|---|
| **Experimental models** | | |
| C57BL/6J (*M. musculus*) (Knock-in) | The University of Michigan Molecular Genetics Core | |
| HEK293T | ATCC | CRL-3216 |
| Human Fibroblast | Weis Lab (Weis et al, 2024) | |

| Reagent/resource | Reference or source | Identifier or catalog number |
|---|---|---|
| **Recombinant DNA** | | |
| pcDNA5-GFP-IL7R-XbaI mutated minigene construct | Kuyumcu-Martinez Lab (Belanger et al, 2018) | |
| pcDNA5-GFP-SEL1Lexon4 | GenScript Biotech | Fig. 3C |
| **Antibodies** | | |
| Rabbit anti-CD19 | Cell Signaling Technology | 90176S |
| Mouse anti-SATB2 | Abcam | ab51502 |
| Rat anti-CTIP2 | Abcam | ab18465 |
| Alexa Fluor 555-conjugated donkey anti-rabbit IgG | Jackson ImmunoResearch | 711-565-152 |
| Alexa Fluor 488-conjugated goat anti-rat IgG | Invitrogen | A-11006 |
| Alexa Fluor 555-conjugated donkey anti-mouse IgG | Invitrogen | A32773 |
| Mouse anti-HSP90 | Santa Cruz | sc-13119 |
| Rabbit anti-SEL1L | home-made (Zhou et al, 2020) | |
| Rabbit anti-SEL1L | Abcam | ab78298 |
| Rabbit anti-HRD1 | Proteintech | 13473-1 |
| Rabbit anti-CD147 | Proteintech | 11989-1 |
| Rabbit anti-IRE1α | Cell Signaling Technology | 3294 |
| Rabbit anti-ERLEC1 | Abcam | ab181166 |
| Rabbit anti-BiP/GRP94 | Abcam | ab21685 |
| Rabbit anti-PDI | Enzo | ADI-SPA-890 |
| Rabbit anti-PERK | Cell Signaling Technology | 3192 |
| Anti-Rabbit IgG TrueBlot HRP | Rockland | 18-8816-33 |
| Anti-Mouse IgG TrueBlot-HRP | Rockland | 18-8817-31 |
| IgG | Cell Signaling Technology | 2729S |
| ZombieNIR Fixable Viability dye | BioLegend | 423106 |
| Rat anti-mouse CD16/CD32 | eBioscience | Ebio 14-0161-85 |
| Rat anti-CD45 (30-F11), | eBioscience | Ebio48-4031-82 |
| Rat anti-CD45R/B220 (RA3-6B2) | BioLegend | 103208 |
| Rat anti-CD19 (6D5) | BioLegend | 115512 |
| **Oligonucleotides and other sequence-based reagents** | | |
| PCR primers | This study | Appendix Table S1 |
| Sequencing primers | This study | Appendix Table S1 |
| Antisense oligonucleotides | This study | Appendix Table S1 |
| **Chemicals, enzymes and other reagents** | | |
| Tunicamycin | Tocris | 3516 |
| PBS | Corning | 21-040-CV |
| 10% Neutral Buffered Formalin | VWR® Histology Reagents | 89370-094 |
| Sucrose | Sigma-Aldrich | SO389 |
| Tissue-Tek® O.C.T. Compound | Sakura Finetek USA Inc | 4583 |
| Normal Donkey Serum | Jackson ImmunoResearch | 017-000-121 |
| Tween-20 | Sigma-Aldrich | P1379-6X1L |
| ProLong™ Diamond Antifade Mountant with DAPI | Fisher Scientific | P36971 |
| TRI Reagent and BCP phase separation reagent | Molecular Research Center | TR 118 |
| Protease inhibitor | Sigma-Aldrich | P8340 |
| DTT | Fisher Scientific | BP172-5 |
| Phosphatase inhibitor cocktail | Bio Basic | PL017 |
| Bio-Rad Protein Assay Dye | Bio-Rad | 5000006 |
| Phos-tag™ Acrylamide | Wako Chemicals | 304-93521 |
| Thapsigargin | Tocris | 1138 |
| EndoH | ACROBiosystems | ENH-S5116 |
| PNGase F | New England BioLabs | P0704L |
| Endo-Porter | Morpholino Gene Tool, LLC | OT-EP-PEG-1 |
| 16% Paraformaldehyde Aqueous Solution, EM Grade | Electron Microscopy Sciences | 15710 |
| EDTA, 0.5 M, pH 8.0, Molecular Biology Grade, DEPC-Treated | Sigma-Aldrich | 324506 |
| Protein A agarose beads | Invitrogen | 15918-014 |
| 30% Acrylamide/Bis Solution, 29:1 | Bio-Rad | 1610156 |
| 40% Acrylamide/Bis Solution, 19:1 | Bio-Rad | 1610144 |
| HotStart Taq 2X PCR Master Mix with Dye | ABclonal | RK20605 |
| Agarose (Low-EEO/Multi-Purpose/Molecular Biology Grade) | Fisher BioReagents | BP160-500 |
| Rabbit TrueBlot®: Anti-Rabbit IgG HRP | Rockland | 18-8816-33 |
| SuperScript™ III Reverse Transcriptase | Invitrogen | 18080093 |
| RNaseOUT™ Recombinant Ribonuclease Inhibitor | Invitrogen | 10777019 |
| Methanol (Certified ACS), Fisher Chemical | Fisher Scientific | A412-4 |
| Gibco™ DMEM, high glucose, pyruvate | Gibco | 11-995-081 |
| Trypsin-EDTA (0.25%), phenol red | Gibco | 25200114 |
| MycoStrip™ - Mycoplasma Detection Kit | Invivogen | rep-mys-20 |
| **Software** | | |
| ImageJ FIJI software | | |
| FCS Express 7 Research | De Novo Software | |
| Image Lab | Bio-Rad | |
| SnapGene | SnapGene | |

| Reagent/resource | Reference or source | Identifier or catalog number |
|---|---|---|
| **Other** | | |
| Leica DMi8 THUNDER Imager | Leica | |
| CBS Scientific Adjustable Height Vertical Gel System | C.B.S. Scientific | |
| Cytek Aurora Northern Lights Spectral Flow Cytometer | Cytek Biosciences | |
| Bio-Rad Electrophoresis Gel System | Bio-Rad | |
| Bio-Rad ChemiDoc Imaging Systems | Bio-Rad | |
| NanoDrop™ One/OneC Microvolume UV-Vis Spectrophotometer | Thermo Scientific | |

## Mice

The *SEL1L^{C141Y}* knock-in (KI) mice (human SEL1L p.Cys141 is equivalent to mouse SEL1L p.Cys137) were generated at the University of Michigan Molecular Genetics Core using the CRISPR-Cas9 technology. The two single guide RNAs (sgRNAs) were designed by using computer algorithm (http://crispor.tefor.net), targeting mouse genome *SEL1L* exon 4 where the mutation is located: sgRNA1: 5'-ATGAGTGCACCTCAGACGGG-3'; sgRNA2: 5'-AAGTGCG-TATTGTTCAAGTG-3'. The sgRNAs were synthesized using the Synthego sgRNA synthesis Kit per manufacturer's protocol, tested and confirmed in fertilized eggs. A donor DNA carrying mouse *Sel1L* cDNA 410 G > A mutation and additional silent mutations was designed to mediate homology-directed repair (HDR): 5'-CACGGG-GAGCCCTGCCACTTCCCTTTCCTTTTCCTGGATAAGGAGTAT GATGAGTACACATCCGATGGAAGAGAAGATGGCAGACTGT GGTGTGCCACAACCTATGACTACAAGACAGATGAGAAGTG GGGCTTCTGCGAAAGTGCGTATTGTTCAAGTGCACGCCCTG TGCTTTAGGGCAGCATTTGGAAGGCATTTTC-3'. The donor DNA was synthesized as Ultramer dsDNA by Integrated DNA Technologies, Inc. A mixture of Cas9 protein (Sigma), sgRNAs, and donor DNA was microinjected into fertilized mouse eggs on the C57BL6/J background. The injected zygotes were then transferred into pseudopregnant females. The existence of desired mutations was further confirmed by Sanger sequencing in three independent founders, which were derived from three separate embryo injections. The founders were then bred separately to WT C57BL6/J mice to obtain F1 heterozygous *SEL1L^{C141Y}* heterozygous mice. F1 heterozygous *SEL1L^{C141Y}* mice were inter-crossed to generate homozygous *SEL1L^{C141Y}* KI mice and its WT and heterozygous littermates.

Animals (or samples) were randomly assigned to experimental groups using a computer-generated randomization list. Group allocation was performed by an investigator not involved in subsequent data collection. All outcome assessments and analyses were conducted under blinded conditions, with investigators unaware of group identities until the completion of data analysis.

For survival curve, pregnant females and litters were checked every 4 h. Pup tails were collected for genotyping. Age- and gender-matched littermates were maintained in a temperature-controlled room on a 12-h light/dark cycle and used in all studies. For in vivo ER stress induction, 12-week-old male WT mice were i.p. injected with DMSO or tunicamycin at 1 mg/kg for 4 h.

All animal procedures were approved by the Institutional Animal Care and Use Committee of the University of Michigan Medical School (PRO00010658), and University of Virginia Medical School (#4459) and in accordance with the National Institutes of Health (NIH) guidelines.

## Human liver tissues

Deidentified frozen normal human liver tissue was acquired through Biorepository and Tissue Research Facility Core at the University of Virginia.

## Genotyping, PCR and sequencing

Mice were routinely genotyped using PCR of genomic DNA samples obtained from tails or ears with the following primer pairs (Appendix Table S1):

*Sel1L^{C141Y}* allele for *SEL1L^{C141Y}* KI mice (Fig. EV1B,C):
F: 5'-AGTACACATCCGATGGAAGAGAAG-3';
R: 5'-GAAAATGCCTTCCAAATGCTGC-3';
*Sel1L* wild-type allele (Fig. EV1B,C):
F: 5'-AGTGCACCTCAGACGGGAGGG-3';
R: 5'-GAAAATGCCTTCCAAATGCTGC-3'.

Mouse tail or ear genomic DNA sample or cell genomic DNA was used for Sanger sequencing analysis by using following primers for PCR and sequencing:

Sequencing for *mSel1L* genomic DNA Exon 4 and intron 4 (Figs. 1C and 3A):
F: 5'-CTTAAGAACTCAAAGTCTACACTAAGTCT-3';
R: 5'-CAGCTTGCCTCAAGGGTTTACAGAA-3'.

## Histology and immunofluorescence

Adult mice were anesthetized and perfused with 20 ml of PBS followed by 40 ml of 10% neutral buffer formalin for fixation. Tissues were dissected out and fixed overnight in 10% neutral buffer formalin at 4 °C. For hematoxylin and eosin (H&E) staining, samples were dehydrated, embedded in paraffin, and stained at the Research Histology Core at the University of Virginia School of Medicine. Images were captured using Aperio ScanScope CS System and Aperio ImageScope software.

For immunofluorescence staining, postnatal P0 pups were anesthetized with hypothermia. Then, pup spleen and brain were dissected followed by overnight fixation in 5 ml 10% neutral buffer formalin under 4 °C. The tissues were then transferred to 30% sucrose in PBS solution for overnight incubation under 4 °C and then embedded in OCT for cryosection. The tissues were sectioned at 8–15 μM. The samples were blocked with 5% normal donkey serum, 0.3% Tween-20 PBS solution for 1 h and then incubated with primary antibody diluted in 5% normal donkey serum, 0.3% Tween-20 PBS solution for overnight at 4 °C: rabbit anti-CD19 (Cell Signaling Technology, 90176S, 1:100); mouse anti-SATB2 (Abcam, ab51502, 1:20); rat anti-CTIP2 (Abcam, ab18465, 1:100). The samples were washed with PBS for 5 min × 3 times at room temperature, and then incubated with secondary antibody for 1 h at room temperature: Alexa Fluor 555-conjugated donkey anti-rabbit IgG (Jackson ImmunoResearch, 711-565-152, 1:500), Alexa Fluor

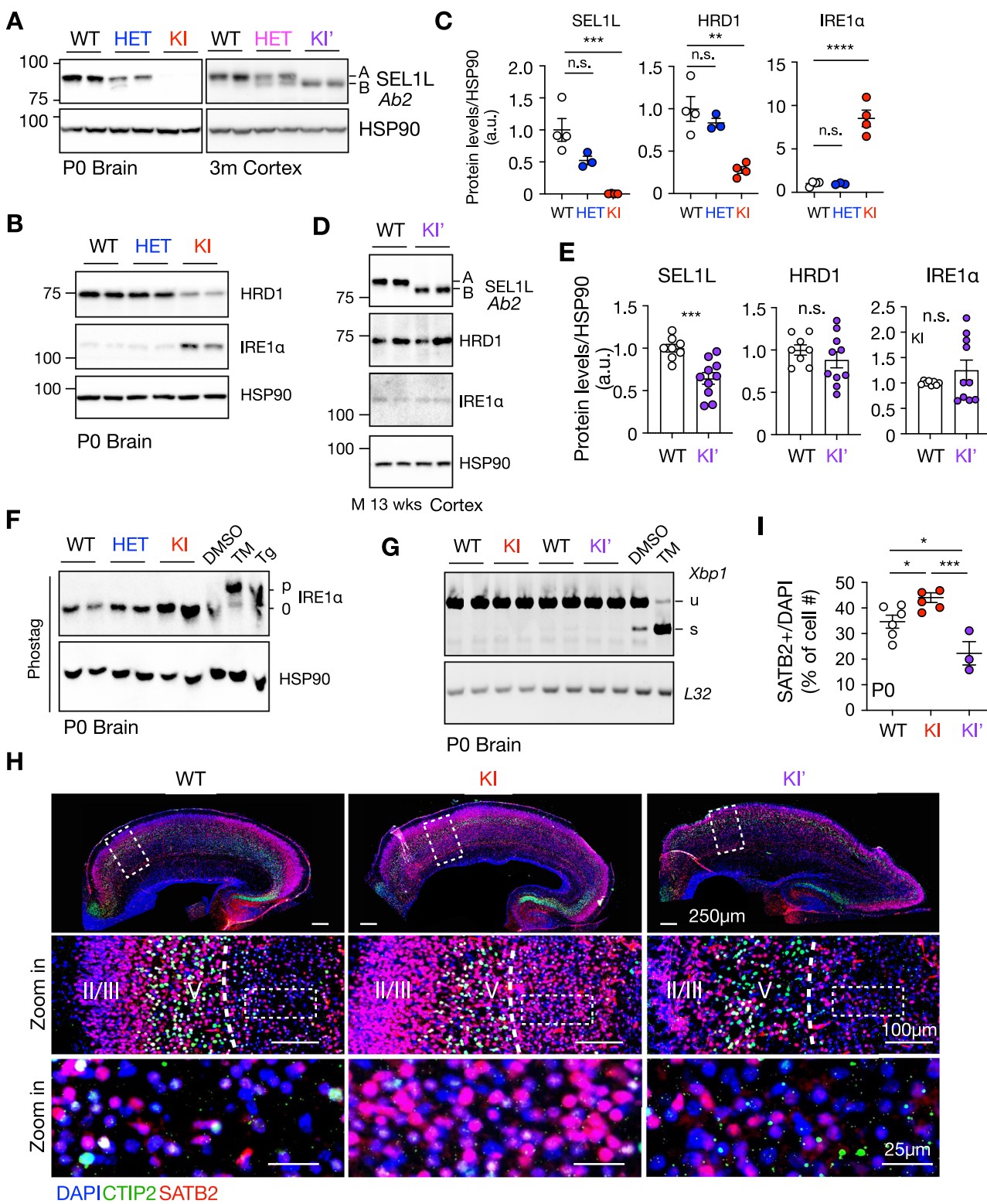

488-conjugated goat anti-rat IgG (Invitrogen, A-11006, 1:500), Alexa Fluor 555-conjugated donkey anti-mouse IgG (Invitrogen, A32773, 1:500). After 5 min × 3 times washing with PBS at room temperature, the slides were mounted in ProLong™ Diamond Antifade Mountant with DAPI (Fisher Scientific P36971). The samples were imaged by Leica DMi8 THUNDER Imager.

## Plasmids

The following plasmids were used in the study: pcDNA5-GFP-IL7R-XbaI mutated minigene construct as described by the Kuyumcu-Martinez Lab (Belanger et al, 2018). pcDNA5-GFP-SEL1Lexon4 were generated by GenScript Biotech, by replacing IL7

**Figure 5. SEL1L C141Y mutation causes ERAD dysfunction and developmental defects in the brain without overt UPR activation.**

(A–C) Western blot analysis of SEL1L protein expression (A), HRD1 and ERAD substrate IRE1α (B) in mouse brains or cortex at indicated ages, with quantification in (C). $n = 3$–4 mice/group. SEL1L: WT vs. HET, $P = 0.057$; WT vs. KI, ***$P = 0.0006$; HRD1: WT vs. HET, $P = 0.52$; WT vs. KI, **$P = 0.0017$; IRE1α: WT vs. HET, $P = 0.99$; WT vs. KI, ****$P < 0.0001$. (D, E) Western blot analysis (D) of ERAD protein and ERAD substrates from WT and KI' mice brain with quantification shown in (E). $n = 3$–4 mice/group. SEL1L: WT vs. KI', ***$P = 0.0009$; HRD1: WT vs. KI', $P = 0.34$; IRE1α: WT vs. KI', $P = 0.28$. (F) Phos-tag gel analysis of IRE1α phosphorylation in p0 WT, HET and KI brains. $n = 3$–4 mice/group. (G) RT-PCR analysis of Xbp1 mRNA splicing of p0 WT, KI, and KI' brains with quantification in Fig. EV4A. $n = 3$–6 mice/group, 3–4 for positive controls. u, unspliced. s, spliced. (H, I) Immunofluorescence (H) of SATB2+ Layer II/III cells and the CTIP2+ Layer V cells in the cortex of p0 WT, KI and KI' mice, with zoom-in region showing SATB2+ cells below Layer V and with quantification shown in (I). $n = 3$–6 mice/group. Green, CTIP2; Red, SATB2; Blue, DAPI. WT vs. KI, *$P = 0.046$; WT vs. KI', *$P = 0.032$; KI vs. KI', ***$P = 0.0007$. Data are represented as means ± SEM. n.s. not significant. *$P < 0.05$; **$P < 0.01$; ***$P < 0.001$, ****$P < 0.0001$, by one-way ANOVA followed by Tukey's post hoc test for (C, I); two-tailed $t$ test for (E). Source data are available online for this figure.

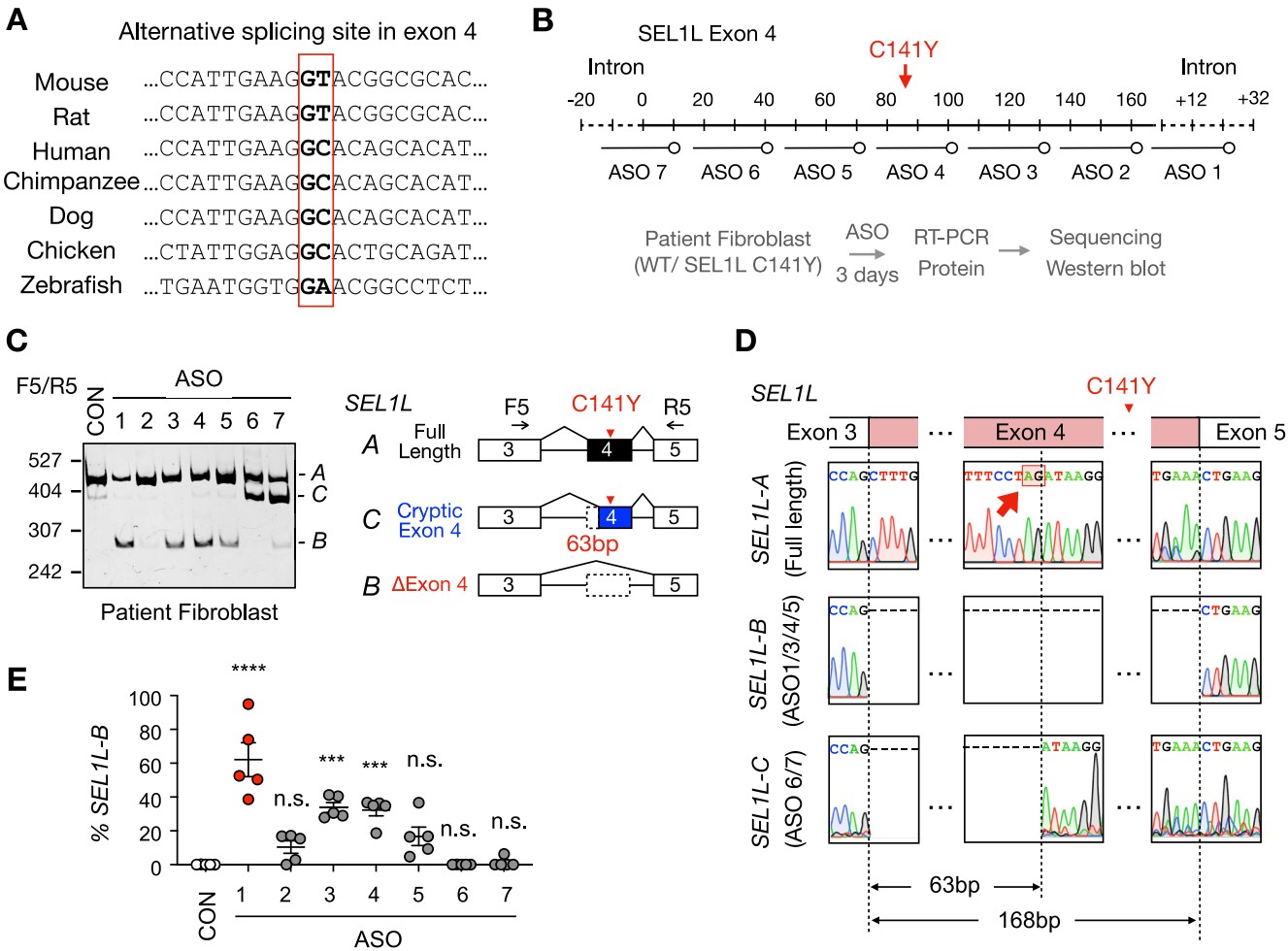

**Figure 6. Screening of ASOs that induce SEL1L exon 4 skipping in patient fibroblasts.**

(A) Sequence alignment of the alternative splice site within Sel1L exon 4 across different species. (B) The experimental design of the ASOs screen targeting SEL1L Exon 4. (C, E) DNA PAGE analysis of ASOs treatment at 10 μM for 3 days in C141Y patient fibroblasts, with a diagram showing corresponding products on the right and with quantification in (E). CON, Control ASO. Red shaded box and arrow indicates the location of SEL1L C141Y mutation. $n = 5$ independent biological replicates. Statistics indicates the comparison between control ASO and ASO 1-7 targeting SEL1L exon 4 and introns 3 and 4. CON vs. 1, ****$P < 0.0001$; CON vs. 2, $P = 0.44$; CON vs. 3, ****$P < 0.0001$; CON vs. 4, ***$P = 0.0001$; CON vs. 5, $P = 0.072$; CON vs. 6, $P > 0.99$; CON vs. 7, $P > 0.99$. (D) Sanger sequencing confirmation of SEL1L-A, -B, and -C bands from (C). Data are represented as means ± SEM. n.s. not significant. **$P < 0.01$; ***$P < 0.001$, ****$P < 0.0001$, by one-way ANOVA followed by Tukey's post hoc test for (E). Source data are available online for this figure.

exon in the pcDNA5-GFP-IL7-XbaI mutated minigene construct with mouse SEL1L exon 4 and the proximal introns (348 bp upstream and 462 bp downstream). Alternative splicing donor site mutation and exon 4 splicing donor site mutation were generated using the pcDNA5-GFP-SEL1Lexon4 minigene construct. All

plasmids were validated by DNA sequencing. The primers used for mutagenesis in Fig. 3D,H are (Appendix Table S1):

Mutagenesis primers for alternative splicing donor site (Mutation 1):

F 5'-CCATTGAAGGCACGGCGCACGGGGA-3'

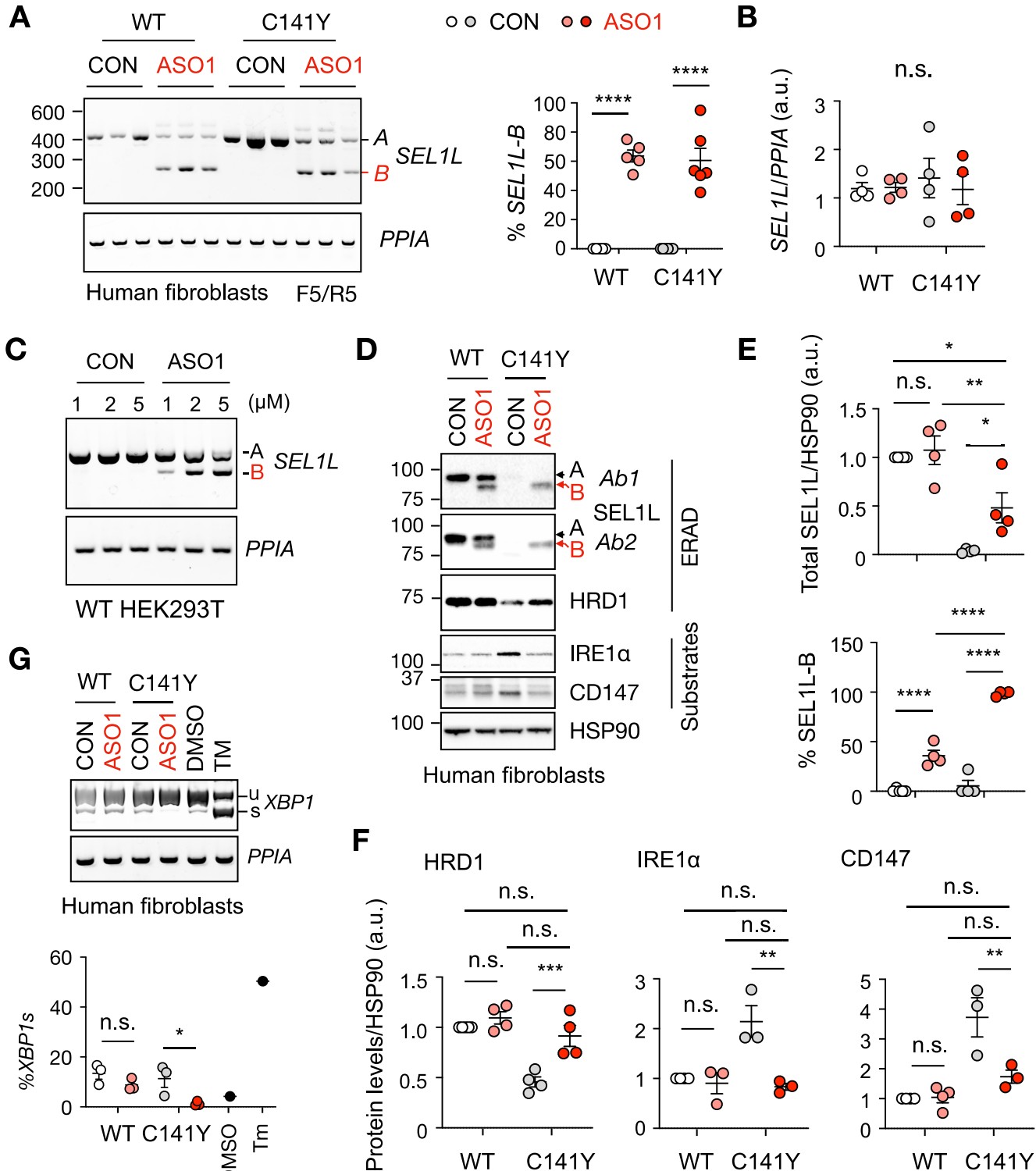

R 5'-GCGCCGTGCCTTCAATGGCAGTCAAGA-3'.
Mutagenesis primers for Exon 4 canonical splicing donor site
mutation (Mutation 2):
F 5'-CTTCTGCGAAAGGCGTATTGTTCAAGTGGGG-3'

R 5'-AACAATACGCCTTTCGCAGAAGCCCCACTTCT-3'
Mutagenesis primers for Exon 4 C137Y mutation (Mutation 3):
F 5'-GTATGATGAGTACACCTCAGACG-3'
R 5'-CGTCTGAGGTGTACTCATCATAC-3'

**Figure 7. ASO-mediated exon skipping rescues ERAD defects in C141Y patient-derived fibroblasts.**

(A) DNA agarose gel electrophoresis analysis of ASO1 treatment at 10 µM for 3 days in human WT and C141Y patient fibroblasts, with quantification shown on the right. CON, Control ASO. $n = 5$–6 independent biological replicates. WT: CON vs. ASO1, ****$P < 0.0001$; C141Y: CON vs. ASO1, ****$P < 0.0001$. (B) Quantification of total SEL1L transcription level in (A). WT: CON vs. ASO1, $P > 0.99$; C141Y: CON vs. ASO1, $P = 0.80$. (C) DNA agarose gel electrophoresis analysis of ASO1 treatment at the indicated concentrations for 24 h in HEK293T cells. (D–F) Western blot analysis (D) of ERAD proteins and ERAD substrates after ASO1 treatment at 10 µM for 3 days in human WT and C141Y patient fibroblasts, with quantification shown in (E, F). Two SEL1L antibodies (AB1, AB2) were used as shown Fig. 2A. $n = 3$–4 independent biological replicates. Total SEL1L: WT(CON) vs. WT(ASO1), $P = 0.95$; C141Y(CON) vs. C141Y(ASO1), *$P = 0.040$; WT(CON) vs. C141Y(ASO1), *$P = 0.011$; WT(ASO1) vs. C141Y (ASO1), **$P = 0062$. %SEL1LB: WT(CON) vs. WT(ASO1), ****$P < 0.0001$; C141Y(CON) vs. C141Y(ASO1), ****$P < 0.0001$; WT(ASO1) vs. C141Y (ASO1), ****$P < 0.0001$. HRD1: WT(CON) vs. WT(ASO1), $P = 0.69$; C141Y(CON) vs. C141Y(ASO1), ***$P = 0.0010$; WT(CON) vs. C141Y(ASO1), $P = 0.75$; WT(ASO1) vs. C141Y (ASO1), $P = 0.23$. IRE1α: WT(CON) vs. WT(ASO1), $P = 0.98$; C141Y(CON) vs. C141Y(ASO1), **$P = 0.0031$; WT(CON) vs. C141Y(ASO1), $P = 0.90$; WT(ASO1) vs. C141Y (ASO1), $P > 0.99$. CD147: WT(CON) vs. WT(ASO1), $P > 0.99$; C141Y(CON) vs. C141Y(ASO1), **$P = 0.0076$; WT(CON) vs. C141Y(ASO1), $P = 0.38$; WT(ASO1) vs. C141Y (ASO1), $P = 0.43$. (G) Electrophoresis analysis of *XBP1* splicing of ASO1 treatment at 10 µM for 3 days in human WT and C141Y patient fibroblasts, with quantification shown below. DNA PAGE gel for the *XBP1* splicing and agarose gel for the internal control *PPIA*. $n = 3$ independent replicates for each group. $n = 1$ for positive controls. u, unspliced. s, spliced. WT: CON vs. ASO1, $P = 0.49$; C141Y: CON vs. ASO1, *$P = 0.021$. Data are represented as means ± SEM. n.s., not significant. *$P < 0.05$; **$P < 0.01$; ***$P < 0.001$; ****$P < 0.0001$, by two-way ANOVA followed by Tukey's multiple-comparisons test for (A, B, E, F), Multiple $t$ tests for (G). Source data are available online for this figure.

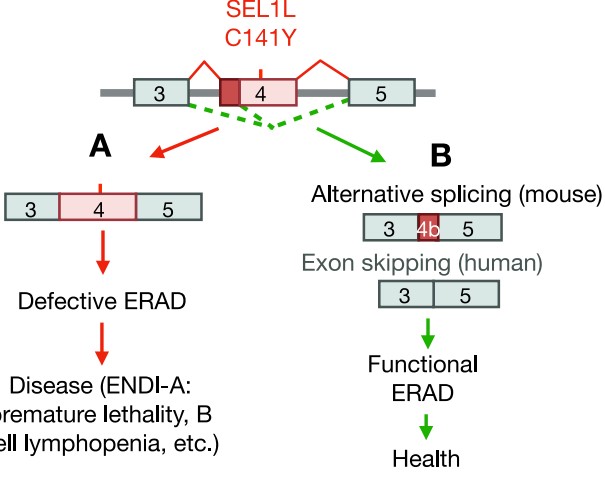

**Figure 8. Schematic model illustrating rescue of the SEL1L C141Y mutation through alternative splicing modulation in mice and humans.**

(A) The SEL1L C141Y homozygous mutation leads to defective ERAD and causes human ENDI-A syndrome, characterized by developmental defects and B cell lymphopenia. (B) In contrast, alternative splicing in mice and ASO-mediated exon skipping in human cells, which bypass the SEL1L C141Y mutation, restore ERAD function and rescue the phenotypes observed in KI mice and patient-derived fibroblasts.

Mutagenesis primers for Exon 4 synonymous mutations (Mutation 4):

F 5'-GATGAGTGCACATCCGATGGAAGAGAAGATGGCAGACTGTGG-3'
R 5'-GCCATCTTCTCTTCCATCGGATGTGCACTCATCATACTCC-3'.

## RNA preparation and RT-PCR

Total RNA was extracted from cell or tissues using TRI Reagent and BCP phase separation reagent (Molecular Research Center, TR 118). For RT-PCR analysis the following primer sequences were used (Appendix Table S1):

*mSEL1L* Full length (F1/R1 used for Fig. 1F):
F: 5'-ATGCAGGTCCGCGTCAGGCTGTCGTTGCTGCT-3';

R: 5'-CTACTGTGGTGGCTGCTGCTCTGG-3';
*hSEL1L* Full length (F1'/R1' used for Fig. 1F):
F: 5'-GCGGCTAGCATGCGGGTCCGGATAGGGCT-3';
R: 5'-GCGAAGCTTTTACTGTGGTGGCTGCTGCTCTG-3';
*mSEL1L* Exon4 for acrylamide gel (F2/R2 used for Figs. 1H,J and EV2B,C):
F: 5'-AGCAAGACCTACGAAGAACT-3';
R: 5'-GAAGGTACCGATATGCTTCTCTCT-3';
*mSEL1L* Exon4 for agarose gel (F3/R3 used for Fig. EV2E,F):
F: 5'-ATGCAGGTCCGCGTCAGGCTGTCGTTGCTGCT-3';
R: 5'-CTACTGTGGTGGCTGCTGCTCTGG -3';
Minigene Exon4 (F4/R4 used for Fig. 3E,I):
F: 5'-TGGTGAGCAAGGGCGAGG-3';
R: 5'- CGTCCTTGAAGAAGATGGTGCG-3';
*hSEL1L* Exon4 (F5/R5 used for Figs. 6C and 7A,B):
F: 5'-GGGGAAAGTGTCACAGAAGATATCAG-3';
R: 5'-GACACTCTCTCCAGGGCTTTG-3';
*mXbp1s*: F: 5'-ACGAGGTTCCAGAGGTGGAG-3'; R: 5'-AAGAGGCAACAGTGTCAGAG-3';
*hXBP1s*: F: 5'-GAATGAAGTGAGGCCAGTGG-3'; R: 5'-ACTGGGTCCTTCTGGGTAGA-3';
*mL32* F: 5'-GAGCAACAAGAAAACCAAGCA-3'; R: 5'- TGCACACAAGCCATCTACTCA-3';
*hPPIA* F: 5'-GGCAAATGCTGGACCCAACACA-3'; R: 5'-TGCTGGTCTTGCCATTCCTGGA-3'.

RT-PCR products were analyzed by 0.8–1.5% agarose electrophoresis or 5% polyacrylamide gel electrophoresis gel using CBS Scientific Adjustable Height Vertical Gel System.

## Western blot and antibodies

Mouse tissues or cells were harvested and snap-frozen in liquid nitrogen. The proteins were extracted by sonication in NP-40 lysis buffer (50 mM Tris-HCl at pH 7.5, 150 mM NaCl, 1% NP-40, 1 mM EDTA) with protease inhibitor (Sigma), DTT (Sigma, 1 mM) and phosphatase inhibitor cocktail (Sigma). Lysates were incubated on ice for 30 min and centrifuged at 16,000 g for 10 min. Supernatants were collected and analyzed for protein concentration using the Bio-Rad Protein Assay Dye (Bio-Rad). In total, 20–50 µg of protein were denatured at 95 °C for 5 min in 5× SDS sample buffer (250 mM Tris-HCl pH 6.8, 10% sodium dodecyl sulfate, 0.05% bromophenol blue, 50% glycerol, and 1.44 M β-mercaptoethanol). Protein was separated on SDS-PAGE, followed by electrophoretic

transfer to PVDF (Fisher Scientific) membrane. The blots were incubated in 2% BSA/Tri-buffered saline tween-20 (TBST) overnight at 4 °C with primary antibodies specifically for: HSP90 (Santa Cruz, #sc-13119, 1:5000), SEL1L (home-made, 1:10,000) (Zhou et al, 2020), SEL1L (Abcam, ab78298, 1:1000), HRD1 (Proteintech, #13473-1, 1:2000), CD147 (Proteintech, #11989-1, 1:3000), IRE1α (Cell Signaling, #3294, 1:2000), ERLEC1 (Abcam, #ab181166, 1:5000), BiP/GRP94 (Abcam, #ab21685, 1:5000), PDI (Enzo, #ADI-SPA-890, 1:5000), PERK (Cell Signaling, #3192, 1:5000). Membranes were washed with TBST and incubated with secondary antibodies, either HRP conjugated (Bio-Rad, 1:10,000), anti-Rabbit IgG TrueBlot HRP (Rockland, #18-8816-33, 1:500) or anti-Mouse IgG TrueBlot-HRP (Rockland, #18-8817-31, 1:500) at room temperature for 1 h for ECL chemiluminescence detection system (Bio-Rad) development. Phos-tag-based western blot analysis was performed as previously described (Yang et al, 2010). Tunicamycin (2 μg/ml for 4 h)- and Thapsigargin (100 nM for 4 h)-treated macrophage cell line RAW 264.7 were used as positive control for the measurement of IRE1α phosphorylation and unfolded protein response (UPR). Band intensity was determined using ImageJ FIJI software.

## EndoH and PNGase F treatment

EndoH (ACROBiosystems, ENH-S5116) and PNGase F (New England BioLabs, P0704L) treatment were performed per manufacturer's protocols. Briefly, following the addition of glycoprotein denaturing buffer (New England BioLabs), tissue lysates were incubated at 100 °C for 10 min, and then digested with EndoH or PNGase F at 37 °C for 1 h. The reaction was stopped by the addition of 5× denaturing sample buffer and boiled at 95 °C for 5 min prior to be loaded onto SDS-PAGE.

## Immunoprecipitation

Mouse livers were homogenized in the NP-40 lysis buffer containing 25 mM Tris-HCl (pH 7.5), 1% NP40, 150 mM NaCl and protease and phosphatase inhibitors, incubated on ice for 1–2 h and centrifuged at 13,000 rpm for 10 min at 4 °C. Equal amount of protein lysates were incubated with the home-made SEL1L antibody (1:10,000) (Zhou et al, 2020) or IgG (Cell Signaling Technology, 2729S) at 4 °C overnight and then followed by protein A agarose beads (Invitrogen 15918-014) for 2 h at 4 °C. Beads were washed three times with lysis buffer, boiled for 5 min in 2× SDS sample buffer and loaded onto SDS-PAGE gels for western blot analysis.

## Antisense oligonucleotide (ASO) design and treatment

Morpholino ASO were either designed by Morpholino Gene Tool, LLC or by the authors, and synthesized using the Morpholino Gene Tool, LLC. Standard control oligos (CCTCTTACCTCAGTTA-CAATTTATA) were purchased from Morpholino Gene Tool, LLC. Morpholino ASO were dissolved with sterilized $H_2O$ at 1 mM. The human fibroblasts or HEK293T cells were routinely check for mycoplasma contamination (Invivogen, #rep-mys-20). The cells were cultured at 80–100% confluency and treated with 10 μM morpholino oligo and 6 μM/mL Endo-Porter (Morpholino Gene Tool, LLC, OT-EP-PEG-1) in 10% serum DMEM medium. The

cells were collected after 1 or 3 days for RNA or protein analysis, respectively. The ASO sequences are (Positions are labeled relative to the first base pair of Exon 4 on the coding strand, Appendix Table S1):

ASO1 ( + 167 bp - +192 bp): TGCCTCCTACTGAGCAATACT TACT

ASO2 ( + 137 bp - +162 bp): AAAAGCCCCACTTTTCATCTG CTTT

ASO3 ( + 107 bp - +132 bp): CATAGGTTGTAGCACACCA-CAG TCT

ASO4 ( + 77 bp - +102 bp): CTTCCCTCCCATCTGATGTATA TTC

ASO5 ( + 47 bp - +86 bp): ACTCCTTATCTAGGAAAAGAAA AGG

ASO6 ( + 17 bp - +42 bp): GGCAGGGCTCCCCATGTGCT GTGCC

ASO7 (-14bp - +11 bp): GGCGGTCAAAGCTGGAATGACAA GA.

## Flow cytometry analysis of PBMCs

Flow cytometric analysis of peripheral blood mononuclear cells (PBMCs) was performed as we described previously (Ji et al, 2016; Ji et al, 2014). In brief, PBMCs were isolated from blood treated with anti-coagulation reagent and red blood cell lysis buffer for 5 min at room temperature, then stained with ZombieNIR Fixable Viability dye (BioLegend 423106) diluted as 1:400 in PBS-$Ca^{2+}$ solution for 10 min at room temperature. Rat anti-mouse CD16/CD32 antibody was then added at 1:100 to PBMCs for blocking at 4 °C for 5 min. Samples were then incubated with 1:100 fluorochrome-conjugated antibody against anti-CD45 (30-F11), anti-CD45R/B220 (RA3-6B2), anti-CD19 (6D5) (BioLegend or eBioscience) at 4 °C for 30 min, followed by 2 washes with flow cytometry (FACS) buffer (5% serum PBS-$Ca^{2+}$ solution) under 4 °C. The samples were then fixed with 2% paraformaldehyde under room temperature for 20 min, followed by two washes. The samples were stored at 4 °C overnight and analyzed on Cytek Aurora Northern Lights Spectral Flow Cytometer at the University of Virginia Flow Cytometry Core. The .fcs file were analyzed on FCS Express 7 Research.

## Statistical analysis

Statistics tests were performed in GraphPad Prism version 10.0 (GraphPad Software). Unless indicated otherwise, values are presented as mean ± standard error of the mean (SEM). All experiments have been repeated at least three times and/or performed with multiple independent biological samples from which representative data are shown. Data were screened for outliers using Robust Outlier detection Test, and exclusions were made only when justified by technical or procedural anomalies. Statistical differences between the groups were compared using the unpaired two-tailed Student's *t* test for two groups, one-way ANOVA followed by Tukey's post hoc test, two-way ANOVA followed by Tukey's multiple-comparisons test for multiple groups, Chi-square test for contingency table, Mantel–Cox test for survival curve. $P < 0.05$ was considered statistically significant.

## Data availability

This study includes no data deposited in external repositories.

The source data of this paper are collected in the following database record: biostudies:S-SCDT-10_1038-S44318-026-00757-5.

## Peer review information

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

## Acknowledgements

We acknowledge Drs. Fowzan Alkuraya, Denisa Weis, Nicola Brunetti-Pierri, and members in the Qi, Sun, Arvan and Kuyumcu-Martinez laboratories for reagents, technical assistance, insightful discussions, and constructive comments on the manuscript. We acknowledge Ellie Sherman in the Jiang lab for the technical assistance on the brain-related experiments. We acknowledge University of Michigan Molecular Genetics Core for the generation of SEL1L C141Y KI mice, the University of Virginia School of Medicine Research Histology Core, Flow Cytometry Core, and Biorepository and Tissue Research Facility Core for their help. The synopsis image was created in BioRender. HHW is supported by Alzheimer's Association Research Fellowship 25AARF-1375486 and K99/R00 NIH Pathway to Independence Award K99HD118580. This work was supported by Additional Ventures Single Ventricle Research Fund 1048010, and in part by 5R01HL157780-03 and 1R01HL175488-01 (MNK-M); R01DK128077, R01DK132068 (SS); Alzheimer's Association 24AARGD-NTF-1187603, 1R01AG089640, R35GM130292 (LQ), and 1R01NS138119 (LQ and SS).

## Author contributions

**Huilun Helen Wang**: Conceptualization; Formal analysis; Investigation; Methodology; Writing—original draft; Writing—review and editing. **Zhihong Wang**: Investigation; Methodology. **Liangguang Leo Lin**: Methodology. **Sunil K Verma**: Methodology. **Weronika Gniadzik**: Methodology. **Hui Wang**: Investigation; Methodology. **Zexin Jason Li**: Visualization. **Emily Whitestone**: Methodology. **Lulu Jiang**: Investigation; Methodology. **Muge N Kuyumcu-Martinez**: Investigation; Methodology. **Shengyi Sun**: Conceptualization; Supervision; Methodology; Writing—review and editing. **Ling Qi**: Conceptualization; Supervision; Funding acquisition; Writing—original draft; Writing—review and editing.

Source data underlying figure panels in this paper may have individual authorship assigned. Where available, figure panel/source data authorship is listed in the following database record: biostudies:S-SCDT-10_1038-S44318-026-00757-5.

## Disclosure and competing interests statement

The authors declare no competing interests.

# Expanded View Figures

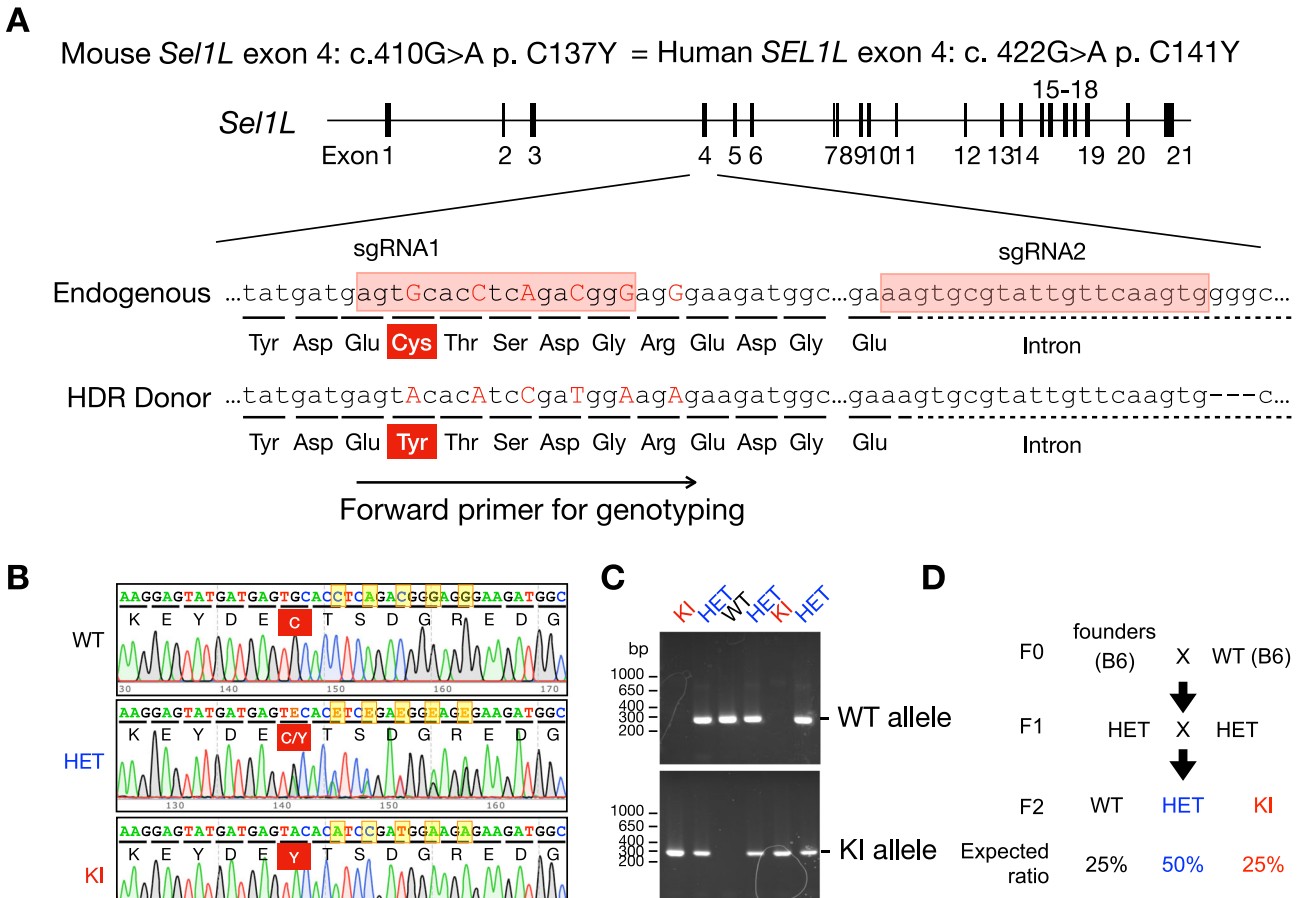

**Figure EV1.　The generation of SEL1L C141Y knock-in (KI) mice (related to Fig. 1).**

(**A**) Diagram of sgRNAs and homology-directed repair (HDR) donor used to generate SEL1L C141Y knock-in mice via CRISPR-Cas9 technology. Silent mutations were introduced to facilitate genotyping, and the "GGG" sequence was deleted in the HDR donor (indicated as "– – –") to disrupt the PAM site. Forward primer for genotyping is indicated below the HDR donor sequence. (**B**) Sanger sequencing confirming the introduction of the SEL1L C141Y mutation along with surrounding silent mutations. E, overlapping peaks. Yellow shades, position of silent mutations. (**C**) DNA agarose gel electrophoresis for genotyping of SEL1L C141Y KI mice. (**D**) Breeding strategy for SEL1L C141Y KI mice. Three established founder lines were bred independently. Source data are available online for this figure.

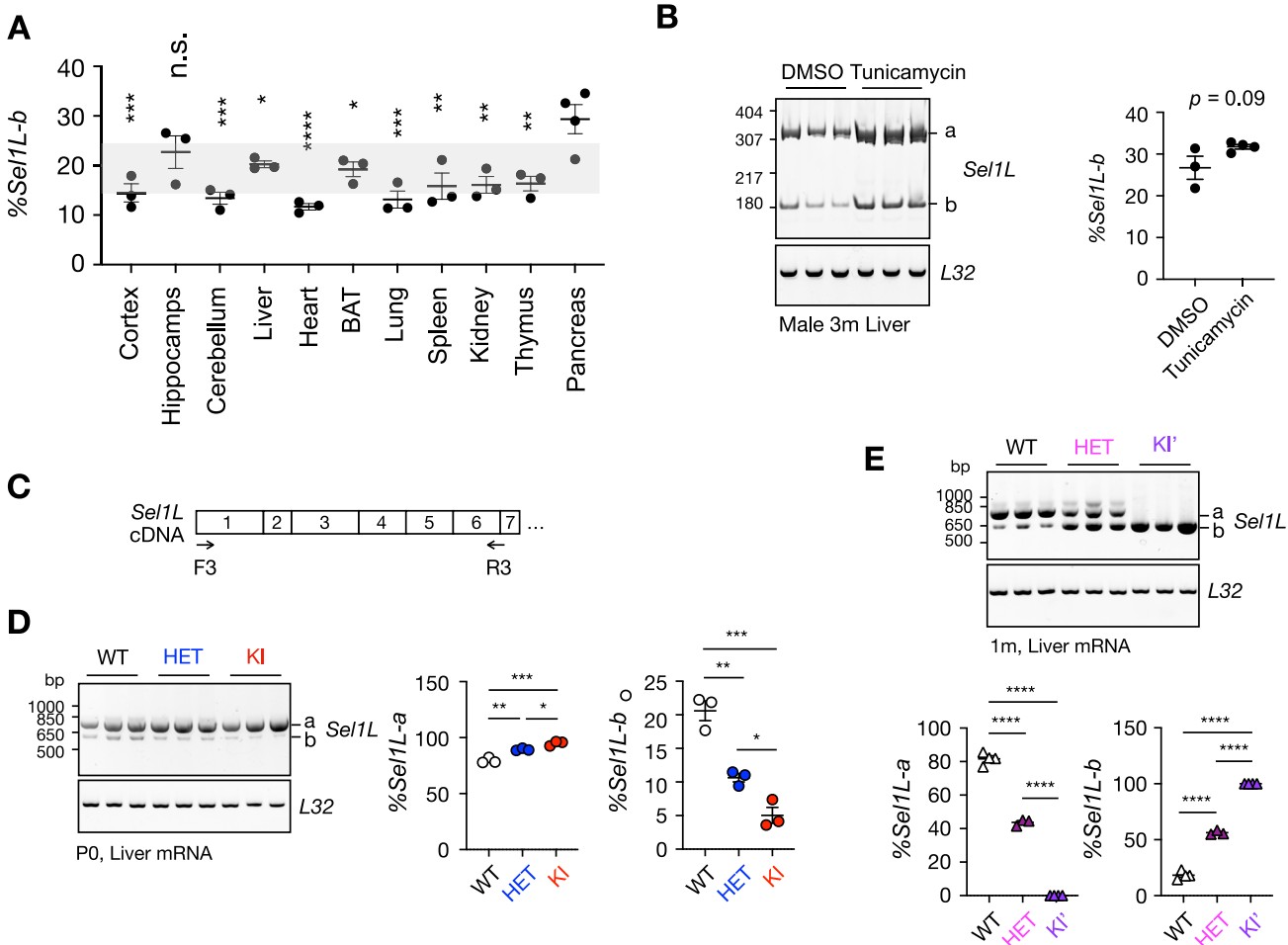

**Figure EV2.  Alternative splicing of *Sel1L* Exon4 under different conditions (related to Fig. 1).**

(A) Quantification of DNA PAGE analysis of *Sel1L* exon 4 splicing in various mouse tissues shown in Fig. 1H. Statistics indicates the comparison between pancreas and other tissues. No statistically significant differences were observed between any of the other tissues. BAT, Brown adipose tissue. $n = 3\text{-}4$ mice/group. (B) DNA polyacrylamide gel electrophoresis (PAGE) analysis of *Sel1L* exon 4 splicing in DMSO or Tunicamycin (Tuni)-treated mouse livers, with quantification shown on the right. $n = 3\text{-}4$ mice/group. DMSO vs. Tunicamycin, $P = 0.092$. (C) Diagram of the primer design for (D, E). (D, E) The agarose electrophoresis of *Sel1L* isoforms in WT, HET, and KI/KI' mice, with quantification shown on the right or below. $n = 3\text{-}4$ mice/group. For (C), %*Sel1L-a*: WT vs. HET, **$P = 0.0020$; WT vs. KI, ***$P = 0.0002$; HET vs. KI, *$P = 0.031$. %*Sel1L-b*: WT vs. HET, **$P = 0.0020$; WT vs. KI, ***$P = 0.0002$; HET vs. KI, *$P = 0.031$. For (D), %*Sel1L-a*: WT vs. HET, ****$P < 0.0001$; WT vs. KI, ****$P < 0.0001$; HET vs. KI, ****$P < 0.0001$. %*Sel1L-b*: WT vs. HET, ****$P < 0.0001$; WT vs. KI, ****$P < 0.0001$; HET vs. KI, ****$P < 0.0001$. Data are represented as means ± SEM. n.s., not significant. *$P < 0.05$; **$P < 0.01$; ***$P < 0.001$, ****$P < 0.0001$, by one-way ANOVA followed by Tukey's post hoc test for (A, D, E), two-tailed *t* test for (B). Source data are available online for this figure.

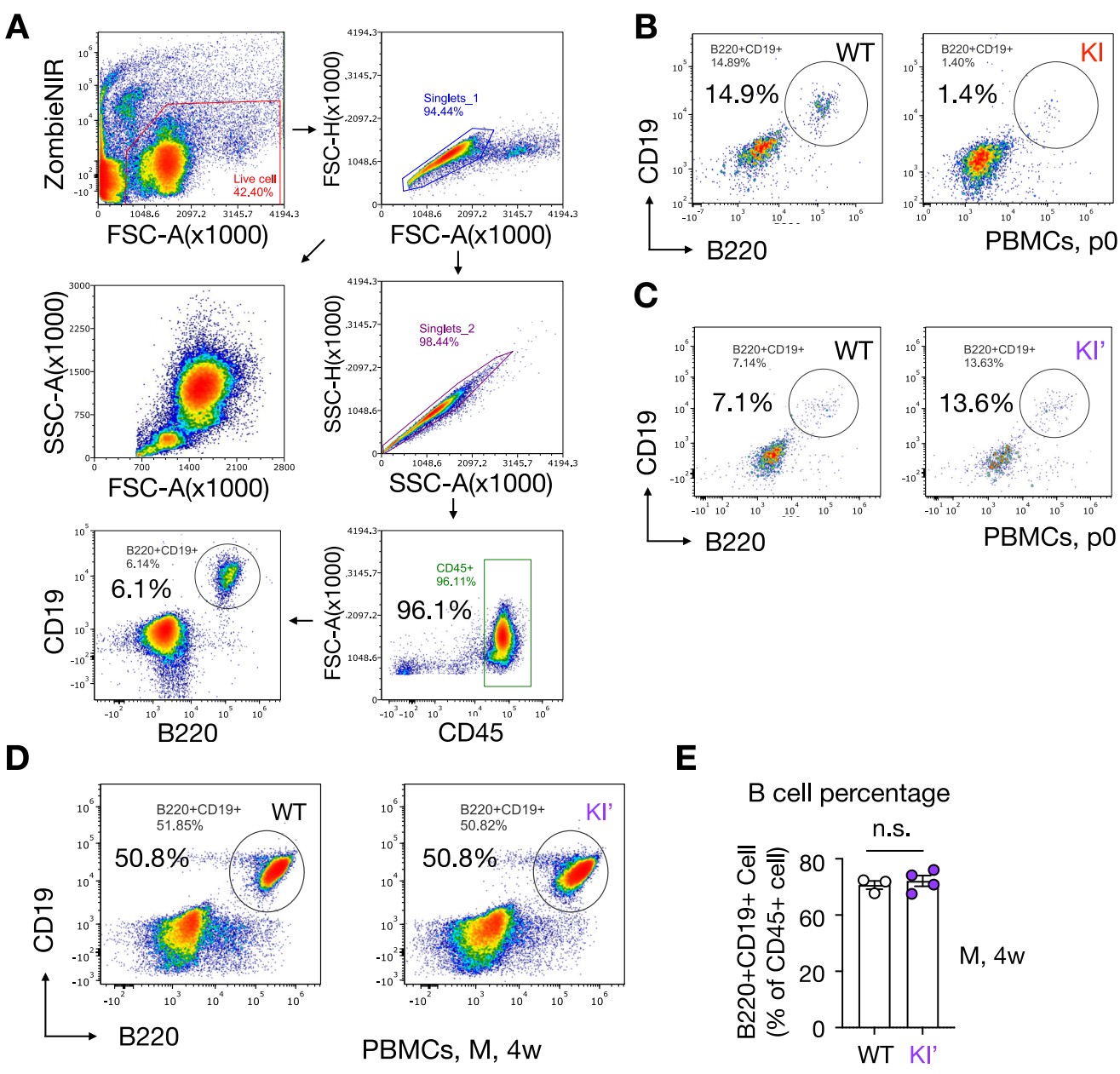

**Figure EV3. B cell deficiency in the circulation of SEL1L C141Y KI mice (related to Fig. 4).**

(A) Gating strategy for flow cytometry. Cells were first gated for live cells using Zombie NIR viability dye, followed by singlet gating to exclude doublets. Live singlet cells were then gated for CD45+ leukocytes, and B cells were identified as CD19+B220+ within the CD45+ population. (B–D) Representative flow cytometry plots showing CD19+B220+ B cells in PBMCs from WT and KI pups at p0 (B), and WT and KI' pups at P0 (C) and 4 weeks of age (D). (E) Quantification of B cells as a percentage of CD45+ cells as shown in (D). $n = 3-4$ mice/group. WT vs. KI', $P = 0.29$. Data are represented as means ± SEM. n.s., not significant using two-tailed Student's $t$ test for (E). Source data are available online for this figure.

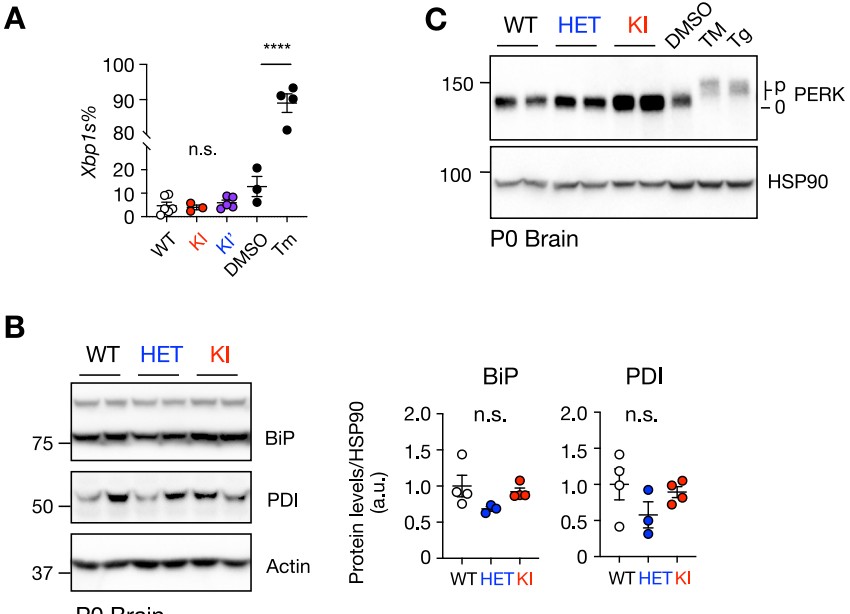

**Figure EV4. SEL1L C141Y KI mice did not cause overt UPR activation (related to Fig. 5).**

(A) Quantification of percentage of XBP1 splicing in Fig. 5G. WT vs. KI, $P > 0.99$; WT vs. KI', $P = 0.99$, DMSO vs. Tm, ****$P < 0.0001$. (B) Western blot analysis of ER chaperones BiP and PDI in P0 WT, HET and KI brains, with quantification shown on the right. $n = 3$–4 mice/group. BiP: WT vs. HET, $P = 0.15$; WT vs. KI, $P = 0.84$; HET vs. KI, $P = 0.31$. PBI: WT vs. HET, $P = 0.25$; WT vs. KI, $P = 0.89$; HET vs. KI, $P = 0.44$. (C) Western blot analysis of PERK in p0 WT, HET and KI brains. $n = 3$–4 mice/group. p, phosphorylated. 0, unphosphorylated. Data are represented as means ± SEM. n.s. not significant; ****$P < 0.0001$ by one-way ANOVA followed by Tukey's post hoc test for (A, B). Source data are available online for this figure.

