## [Peer Review File · The EMBO Journal]

Functional rescue of a disease-linked ERAD pathway mutation via alternative splicing

Huilun Wang, Zhihong Wang, Lianguang Lin, Sunil Verma, Weronika Gniadzik, Hui Wang, Zexin Li, Emily Whitestone, Lulu Jiang, Muge Kuyumcu-Martinez, Shengyi Sun, and Ling Qi

Corresponding author(s): Ling Qi (xvr2hm@virginia.edu) , Huilun Wang (kma8py@virginia.edu), Shengyi Sun (bjk5fz@virginia.edu)

Review Timeline:

Submission Date:	7th Jul 25
Editorial Decision:	4th Sep 25
Revision Received:	20th Nov 25
Editorial Decision:	19th Jan 26
Revision Received:	20th Jan 26
Accepted:	19th Feb 26

Editor: Cornelius Schneider

Transaction Report:

Dear Prof. Qi,

Thank you for submitting your manuscript for consideration by the EMBO Journal. It has now been seen by three referees whose comments are shown below.

As you can see from the reports all three referees think that the reported results are very interesting and that the experiments are generally of high quality. The referees also mention several additional experiments which we agree would enhance the mechanistic insight and overall advance.

Given the referees' positive recommendations, I would like to invite you to submit a revised version of the manuscript, addressing the comments of all three reviewers. I should add that it is EMBO Journal policy to allow only a single round of revision, and acceptance of your manuscript will therefore depend on the completeness of your responses in this revised version.

If you have any additional questions regarding the revisions such as specific experiments or referee comments I would be happy to discuss the revision further via e-mail or videoconferencing.

Thank you for the opportunity to consider your work for publication. I look forward to your revision.

Yours sincerely,

Cornelius Schneider, PhD
Editor
The EMBO Journal
c.schneider@embojournal.org

We realize that it is difficult to revise to a specific deadline. In the interest of protecting the conceptual advance provided by the work, we recommend a revision within 3 months (2nd Dec 2025). Please discuss the revision progress ahead of this time with the editor if you require more time to complete the revisions. Use the link below to submit your revision:

Referee #1:

In this manuscript, the authors have generated a knock-in mouse in Sel1l gene, using Crispr-Cas9, to investigate the molecular mechanisms contributing to ENDI syndrome, an ERAD-associated neurodevelopmental disorder with infantile onset. The knock-in targeted to change Cysteine 141 to Tyrosine in the Sel1L gene. This missense mutation was previously reported in both alleles in patients belonging to a consanguineous family with more severe clinical symptoms, including agammaglobulinemia and premature death (ENDI-I).

Three independent founder lines were generated (Lines A, B, and C) with a confirmed G to A mutation (C141Y). Most of the homozygous knock-in pups derived from lines A and B (named KI) died within 48 hours after birth (5 out of 280 pups survived past 21 days), but homozygous pups from line C (KI') survived at the expected Mendelian ratio. Sequencing of RT-PCR products from the Sel1L transcripts showed a large transcript (Sel1L-a) that includes exon 4 (encoding the C141 residue) and a smaller transcript (Sel1L-b) due to skipping of 150bp within exon 4 due to the use of an upstream 5' splice site within exon 4. Most tissues from WT mice show ~80% Sel1L-a transcripts and ~20% of Sel1L-b transcripts (missing 150bp in exon 4), and KI lines show increased levels of Sel1L-a transcripts, but KI' line shows 100% expression of Sel1L-b transcripts and no expression of Sel1L-a transcripts.

The P0 pup livers show almost no SEL1L protein expression in the KI line, despite the transcript expression similar to that of WT mice. However, KI' livers show a smaller protein, albeit at lower levels (~50%, actual quantification seems a bit lower in the graph Fig.1i, indicate the mean value). The authors also showed reduced HRD1 protein level in KI mice, consistent with a previous report that SEL1L and HRD1 stabilize each other. Authors also showed reduced liver expression of ERLEC1 and increased expression of ERAD substrate IRE α in KI mice but not KI' mice. The authors propose that skipping of 150bp in exon 4 causes the expression of a smaller protein, SEL1L-B, that is stable and partially functional in stabilizing HRD1 and ERAD, which rescues the defect caused by the C141Y mutation.

To investigate the mechanism by which KI' transcripts show increased skipping of exon 4, authors sequenced the genomic region surrounding the Sel1L exon 4. The 5' splice site of exon 4 in KI' was mutated from GT to G- (deletion of T), likely during CRISPR-Cas9 manipulation, which contributed to increased use of the 5' splice site within exon 4 to generate the Sel1L-B transcript in KI' mice. Authors utilized minigene constructs to demonstrate that 5' splice site deletion caused skipping of exon 4 but was not sufficient to cause preferential utilization of internal 5' splice site. Minigene experiments showed that a nonsynonymous (G to A, C141Y) mutation also caused skipping of exon 4 but did not promote the preferential utilization of the internal 5' splice site, yielding the Sel1L-B isoform. However, authors identified five other synonymous mutations near the C141Y site that CRISPR-Cas9 likely created during the knock-in process. Recreating all these mutations together promoted the use of the exon 4 internal 5' splice site in the minigene context.

A detailed characterization of the homozygous KI mice showed reduced mature B cells and defects in brain cortical structures at P0, whereas the KI' mice did not show these phenotypes. Authors correlate these phenotypes with ENDI-I patients' symptoms, which include agammaglobulinemia, intellectual disability, and premature death. The reduced expression of SEL1L in KI mice correlated with reduced mature B cells, and brain cortical structure in these mice, whereas KI' mice showed expression of smaller SEL1L protein, and did not show phenotypes in B cells or cortical brain structures or premature death. Based on these serendipitous results, authors utilized anti-sense oligos (ASOs) to induce skipping of exon 4 in human patient fibroblasts (as internal exon 4 5' splice site is absent in human), which led to expression of a smaller SEL1L protein along with a reduction in IRE1 α , suggesting ASO as a therapeutic approach for ENDI-I patients affected by C141Y mutations.

Major issues:

1. Given that CRISPR-Cas9 has off-target effects. Authors should show rescue of founder lines A and B by ASOs in vivo or cells

derived from these mice. This is critical for valid interpretation of the phenotypes coming from the KI mice are due to targeted KI and not due to CRISPR off-target effect.

2. Authors should also show which aspects of ERAD biology is restored by expression of smaller SEL1L protein lacking the FNII domain, especially when there is minimal UPR response in KI or KI' brains. Whether the smaller protein is being targeted to the ER or elsewhere, interaction with other ERAD proteins HRD1, OS9, etc.

3. Authors should also show RT-PCR from KI and KI' mice tissues, as shown in EV2A. The synonymous and non-synonymous mutation created by CRISPR-Cas9 will affect alternative splicing differently in different tissues, as there is no mention of how alternative splicing is regulated in mouse tissues.

4. It is not indicated why the KI mice don't show the Sel1L-b transcripts, as in the minigene context, the 5 synonymous mutations are sufficient to cause splicing of Sel1L-b-like transcripts. The 5 synonymous mutations near the C141Y region are exclusively in KI' mice or also in KI mice?

Minor issues:

1. Indicate the region being encoded by exon 4 in Figure EV4F.

2. Indicate which ASO is targeting the C141Y region in human fibroblasts.

3. The protein nomenclature should be different for smaller protein generated in mouse (SEL1L-B presumably lacking 50 aa) and human (Δ exon4, lacking 56 aa) induced by partial or full exon 4 skipping, respectively.

4. Significantly is misspelled as "significatntly" in the 2nd paragraph of the last results section.

Referee #2:

In the manuscript 'Functional rescue of a fatal ERAD mutation via alternative splicing', Wang and colleagues report that alternative splicing of SEL1L can ameliorate pathological phenotypes caused by the bi-allelic SEL1L Cys141Tyr mutation. Using several knock-in mouse models, the authors characterize an alternative splicing event resulting in Sel1l-b mRNA and assess the impact of increased levels in the context of a bi-allelic Sel1l C141 mutation on survival, ENDI-like phenotypes, SEL1L/HRD1 expression, ERAD function, and UPR activation. They further evaluate antisense oligonucleotide-mediated exon skipping in patient-derived fibroblasts as a potential therapeutic strategy to counteract SEL1L Cys141Tyr-associated pathology.

The manuscript is well structured, clearly written, and focused. The findings are of broad interest, provide new insight into SEL1L function, and suggest avenues for therapeutic intervention. I have no major concerns and consider the study, in principle, suitable for publication in EMBO J. Below suggestions to further strengthen the work before publication:

Minor

To increase the value for basic research and to substantiate the finding of functionality of the short SEL1L isoform and potentially the role of the FNII domain, I suggest to add a comparative global proteomic mass-spectrometry dataset from patient-derived fibroblasts {plus minus} ASO1, ideally benchmarked against family-derived wild-type fibroblasts.

As an alternative or complement, comparative global proteomes from WT vs. KI vs. KI' liver or brain tissue would be highly informative.

Nonessential

- A mass-spectrometry interactome comparing long vs. short SEL1L would be valuable for the same reasons. However, depending on tool availability (e.g. functionality of the abcam antibody in IP), this may fall outside the scope of the present study.
- Throughout the manuscript, I suggest to avoid presenting IRE1 solely as an ERAD substrate. Because readers less familiar to the system may not be able to grasp the full picture. Better explicitly link IRE1 also to the UPR (e.g., 'the UPR sensor and ERAD substrate, IRE1') and cite an appropriate reference.

Reviewer expertise: cell biology with a focus on ER quality control; limited expertise in in-vivo mouse diagnostics and splicing mechanisms.

Referee #3:

The manuscript by Wang H. et al. investigates the pathogenic mechanism of ERAD-associated neurodevelopmental disorders with onset in infancy (ENDI) and explores potential therapeutic strategies. Previous work from this group linked recessive mutations in SEL1L and HRD1 to ENDI. To model the disease, the authors used CRISPR to introduce a C141Y point mutation in SEL1L in mice. Most homozygous mutants died shortly after birth, but one line survived to adulthood without obvious defects. Mechanistic analysis revealed that this line carried an additional splice-donor mutation, producing a truncated SEL1L protein lacking a fibronectin domain yet retaining ERAD activity. This truncated protein rescued the developmental defects otherwise caused by SEL1L loss. The authors further validated these findings in patient-derived fibroblasts carrying the same mutation.

Overall, this study provides strong evidence that the C141Y mutation destabilizes SEL1L, impairing ERAD and leading to developmental defects. The discovery that an alternatively spliced SEL1L variant retains function and rescues the phenotype offers important new insight and suggests a potential therapeutic avenue for this devastating disease. I have only a few minor

suggestions for revision.

Specific points:

1. The alternative splicing-based mechanism described here appears to be unique to this case, as most truncated proteins are non-functional. In the abstract, the authors should avoid overstating its broader therapeutic potential for other genetic disorders.
2. Please spell out "ERAD-associated neurodevelopmental disorders with onset in infancy (ENDI)" at its first mention.
3. Page 3: replace "largely uncoupled from ER stress response" with "largely independent of the ER stress response."
4. Next paragraph: "bi-allelic SEL1L and HRD1 variants" should be revised to "bi-allelic SEL1L or HRD1 variants." The same correction applies to the following sentence.
5. Page 4: the phrase "three independent founder lines" is unclear. Does this refer to three parental lines of distinct genetic backgrounds, or to three lines of the same background?
6. The sentence "Given their shared phenotypes..." is ambiguous. Since the prior sentence refers to heterozygous mice from all lines, does "their" refer to these heterozygous lines?
7. Figure 1H: truncated SEL1L levels appear comparable to full-length SEL1L in wild-type animals, yet the quantification in Figure 1I indicates only partial rescue. Please clarify.
8. Figure 2C: the intron labels are confusing. Why are they designated as 3' or 5'?
9. Figure 4B: this figure is not mentioned in the main text.
10. Page 9: the sentence "although disulfide bonds within the FNII domain are critical for SEL1L function" could be clarified as "although disulfide bonds within the FNII domain are critical for SEL1L stability when the FNII domain is present."
11. Citation issues:
Lilley & Ploegh (2004) does not discuss SEL1L. It was entirely focused on Derlin-1.
Oda et al. (2006) focuses on Derlins, not SEL1L.
For SEL1L's role in HRD1 stability, the authors should cite PMID: 23867461 (see Figure 5).
For SEL1L's role in substrate recruitment, they should cite Daniel Hebert's work (PMID: 19524542).
The reference "Wang HH" should be updated, as the work is now published.

We sincerely thank the reviewers for the insightful comments and constructive suggestions. We have carefully considered each point raised and revised the manuscript accordingly. Below, we provide a detailed, point-by-point response, highlighting the changes made and clarifying our rationale where necessary.

Referee #1:

In this manuscript, the authors have generated a knock-in mouse in Sel1l gene, using Crispr-Cas9, to investigate the molecular mechanisms contributing to ENDI syndrome, an ERAD-associated neurodevelopmental disorder with infantile onset. The knock-in targeted to change Cysteine 141 to Tyrosine in the Sel1L gene. This missense mutation was previously reported in both alleles in patients belonging to a consanguineous family with more severe clinical symptoms, including agammaglobulinemia and premature death (ENDI-I).

Three independent founder lines were generated (Lines A, B, and C) with a confirmed G to A mutation (C141Y). Most of the homozygous knock-in pups derived from lines A and B (named KI) died within 48 hours after birth (5 out of 280 pups survived past 21 days), but homozygous pups from line C (KI') survived at the expected Mendelian ratio. Sequencing of RT-PCR products from the Sel1L transcripts showed a large transcript (Sel1L-a) that includes exon 4 (encoding the C141 residue) and a smaller transcript (Sel1L-b) due to skipping of 150bp within exon 4 due to the use of an upstream 5' splice site within exon 4. Most tissues from WT mice show ~80% Sel1L-a transcripts and ~20% of Sel1L-b transcripts (missing 150bp in exon 4), and KI lines show increased levels of Sel1L-a transcripts, but KI' line shows 100% expression of Sel1L-b transcripts and no expression of Sel1L-a transcripts.

The P0 pup livers show almost no SEL1L protein expression in the KI line, despite the transcript expression similar to that of WT mice. However, KI' livers show a smaller protein, albeit at lower levels (~50%, actual quantification seems a bit lower in the graph Fig.1i, indicate the mean value). The authors also showed reduced HRD1 protein level in KI mice, consistent with a previous report that SEL1L and HRD1 stabilize each other. Authors also showed reduced liver expression of ERLEC1 and increased expression of ERAD substrate IRE α in KI mice but not KI' mice. The authors propose that skipping of 150bp in exon 4 causes the expression of a smaller protein, SEL1L-B, that is stable and partially functional in stabilizing HRD1 and ERAD, which rescues the defect caused by the C141Y mutation.

To investigate the mechanism by which KI' transcripts show increased skipping of exon 4, authors sequenced the genomic region surrounding the Sel1L exon 4. The 5' splice site of exon 4 in KI' was mutated from GT to G- (deletion of T), likely during CRISPR-Cas9 manipulation, which contributed to increased use of the 5' splice site within exon 4 to generate the Sel1L-B transcript in KI' mice. Authors utilized minigene constructs to demonstrate that 5' splice site deletion caused skipping of exon 4 but was not sufficient to cause preferential utilization of internal 5' splice site. Minigene experiments showed that a nonsynonymous (G to A, C141Y) mutation also caused skipping of exon 4 but did not promote the preferential utilization of the internal 5' splice site, yielding the Sel1L-B isoform. However, authors identified five other synonymous mutations near the C141Y site that CRISPR-Cas9 likely created during the knock-in process. Recreating all these mutations together promoted the use of the exon 4 internal 5' splice site in the minigene context.

A detailed characterization of the homozygous KI mice showed reduced mature B cells and defects in brain cortical structures at P0, whereas the KI' mice did not show these phenotypes. Authors correlate these phenotypes with ENDI-I patients' symptoms, which include agammaglobulinemia, intellectual disability, and premature death. The reduced expression of SEL1L in KI mice correlated with reduced mature B cells, and brain cortical structure in these mice, whereas KI' mice showed expression of smaller SEL1L protein, and did not show phenotypes in B cells or cortical brain structures or premature death. Based on these serendipitous results, authors utilized anti-sense oligos (ASOs) to induce skipping of exon 4 in human patient fibroblasts (as internal exon 4 5' splice site is absent in human), which led to expression of a smaller SEL1L protein along with a reduction in IRE1 α , suggesting ASO as a therapeutic approach for ENDI-I patients affected by C141Y mutations.

Major issues:

1. Given that CRISPR-Cas9 has off-target effects. Authors should show rescue of founder lines A and B by ASOs in vivo or cells derived from these mice. This is critical for valid interpretation of the phenotypes coming from the KI mice are due to targeted KI and not due to CRISPR off-target effect.

We thank the reviewer for the suggestion. Two independent KI founder lines, A and B, were bred separately and exhibited similar phenotypes, indicating that the observed phenotypes are unlikely to result from potential CRISPR off-target effects. Per reviewer's suggestion, we now have performed ASO screening using mouse embryonic fibroblast (MEF) cells derived from founder line A, targeting *Sel1L* Exon4 (**Response Figure 1A**). We found that while multiple ASOs could effectively induced the alternative splicing in WT MEF cells, producing full length isoform *Sel1L-a*, alternative spliced isoform *Sel1L-b*, and Exon 4 skipping isoform *Sel1L-c* (**Response Figure 1B-C**), KI MEFs were resistant to ASO treatment compared to WT MEFs (**Response Figure 1B**). When ASO2 and ASO4 was used in combination, exon-skipping efficiency was enhanced at both *Sel1L-b* mRNA (**Response Figure 1B,D**) and SEL1L-B protein (**Response Figure 1E-F**) levels. This treatment stabilized HRD1 and reduced the accumulation of IRE1 α (**Response Figure 1E and 1G**), thereby rescuing ERAD function. As these data are part of our ongoing effort to investigate the therapeutic efficacy of ASOs *in vitro* and *in vivo*, we consider them beyond the scope of the current study and plan to include them in our next manuscript.

Figure for referee with unpublished data and its description has been removed upon request by the authors.

2. Authors should also show which aspects of ERAD biology is restored by expression of smaller SEL1L protein lacking the FNII domain, especially when there is minimal UPR response in KI or KI' brains.

Whether the smaller protein is being targeted to the ER or elsewhere, interaction with other ERAD proteins HRD1, OS9, etc.

We thank the reviewer for this helpful comment. We have now examined the subcellular localization and protein-protein interactions of truncated SEL1L-B with other ERAD components using Endo H digestion and immunoprecipitation analyses (**Response Figure 2**). The Endo H assay is an enzymatic de-glycosylation test that distinguishes high-mannose (Endo H-sensitive) from complex (Endo H-resistant) N-linked glycans, thereby assessing glycoprotein maturation and subcellular trafficking. PNGase F, which removes almost all types of N-linked glycans, was used as a control to confirm complete de-glycosylation. We found that both isoforms of SEL1L (A- and B-) were Endo H sensitive, similar size to those observed following PNGase F digestion, indicating that both forms are retained within the ER (**Response Figure 2A**). Furthermore, immunoprecipitation experiments showed that SEL1L-B continues to interact with key ERAD components, including HRD1 and OS9 (**Response Figure 2B**). Together, these results suggest that SEL1L-B is localized to the ER and remains functionally comparable to full-length SEL1L-A. These data are now presented in **Figure 2** of the revised manuscript.

Response Figure 2. SEL1L-B is localized to ER and interact with other ERAD components. (A) Representative Western blot of SEL1L-A or SEL1L-B protein from postnatal day 0 liver of WT or KI' mouse following with EndoH or PNGase F treatment. s and s', sensitive band. (B) Immunoprecipitation of SEL1L-A or SEL1L-B protein from postnatal day 0 liver of WT or KI' mouse to test their interactions with components of the ERAD complex. -, no enzyme treatment. E, EndoH treatment. P, PNGase F treatment. n = 3 mice/group. Data are represented as means ± SEM. n.s., not significant, by two-tailed t test for (B).

3. Authors should also show RT-PCR from KI and KI' mice tissues, as shown in EV2A. The synonymous and non-synonymous mutation created by CRISPR-Cas9 will affect alternative splicing differently in different tissues, as there is no mention of how alternative splicing is regulated in mouse tissues.

We agree with the reviewer on this point. We have now analyzed the alternative splicing of SEL1L across multiple tissues (including brain, liver, thymus and pancreas) from both KI and KI' mice (**Response Figure 3**). Our results show that the alternative splicing pattern is consistent among the tissues examined, with no major tissue-specific differences observed among WT, KI, and KI' mice. These findings indicate that the observed splicing changes are intrinsic to the mutation itself rather than secondary to tissue context. These data are now presented in **Figure 1K-L** of the revised manuscript.

Response Figure 3. Alternative splicing of Sel1L across different tissues from KI and KI' mice. (A-B) DNA polyacrylamide gel electrophoresis (PAGE) analysis (A) of Sel1L exon 4 splicing in the brain, liver, thymus and pancreas from WT, KI and KI' p0 pups, with quantification shown in (B). Statistics indicates the comparison between the same tissue type of WT vs. KI or WT vs. KI'. n = 3

mice/group. Data are represented as means ± SEM. n.s., not significant. *p < 0.05; ****p < 0.0001, by Two-way ANOVA followed by Dunnett's multiple comparisons test for (B).

4. It is not indicated why the KI mice don't show the Sel1L-b transcripts, as in the minigene context, the 5 synonymous mutations are sufficient to cause splicing of Sel1L-b-like transcripts. The 5 synonymous mutations near the C141Y region are exclusively in KI' mice or also in KI mice?

We appreciate the reviewer's insightful comments. The five synonymous mutations near the C141Y region are present in both KI and KI' mice. *In vivo*, KI mice predominantly express the *Sel1L-a* transcript (approximately 95%), whereas KI' mice exclusively express the *Sel1L-b* transcript (**Response Fig. 4A-B**).

To investigate how this difference in splicing regulation occurs between KI and KI' mice, we introduced the *Sel1L* C141Y mutation (#3) and the synonymous mutations (#4) into either the wild-type (WT) minigene construct or a construct carrying a mutation in the canonical splice donor site (#2) (**Response Fig. 4C-D**). The C141Y mutation did not affect splicing. However, the synonymous mutations reduced usage of alternative splice donor when the canonical site was intact, but enhanced usage of the alternative splice donor when the canonical site disrupted (**Response Fig. 4E-F**). These results align with our *in vivo* findings, in which the KI and KI' mice predominantly use the canonical splice donor and the internal alternative donor, respectively.

This data demonstrated that the synonymous mutations introduced in the KI mice participate in splicing regulation. Sequence analysis suggests that these synonymous changes may have inadvertently created an exonic splicing enhancer (ESE) motif recognized by Serine/Arginine-rich (SR) proteins (Long & Caceres, 2009). In the wild-type sequence (AGT GCA CCT CAG ACG GGA GGG AA), the purine-rich region (GGA GGG AA) does not strongly resemble canonical ESE consensus motifs typically bound by SR proteins. In contrast, the synonymous mutations in the KI Line (AGT ACA CAT CCG ATG GAA GAG AA) introduces a core element (GAA GAG AA) that closely matches well-characterized ESE motifs such as GAAGAA or GAAGAAG (Tacke & Manley, 1995), which are recognized by SR proteins including SRSF1 and TRA2-β (Long & Caceres, 2009; Shepard & Hertel, 2009). This likely enhances exon recognition and promotes exon inclusion in the presence of the synonymous mutations (Ibrahim *et al*, 2005). Further investigation will be needed to confirm the functional ESE motif and to identify the specific SR proteins involved in this splicing regulation. This data is now shown in **Figure 3I-J** in the revised manuscript.

Response Figure 4. The mutation of major splicing donor enables the usage of an internal alternative splice donor. (A) The diagram of the Sel1L isoforms and the primer design for **(B)**. **(B)** DNA polyacrylamide gel electrophoresis (PAGE) analysis of Sel1L exon 4 isoforms in mouse livers at indicated ages with quantification shown on the right. n = 3-4 mice/group. **(C-D)** Diagrams of the indicated mutations in the minigene reporter construct. **(E-F)** DNA PAGE analysis of the WT and mutated minigene reporters as indicated by using primer pair F4/R4, with quantification in **(F)**. n = 3-6 independent biological replicates. Data are represented as means \pm SEM. n.s., not significant. *p < 0.05; **p < 0.01; ***p < 0.001, ****p < 0.0001, by Two-way ANOVA followed by Tukey's multiple-comparisons test for **(B, F)**.

Minor issues:

1. Indicate the region being encoded by exon 4 in Figure EV4F.

We have now indicated the region encoded by exon 4 in **Figure 2A** (previous **Figure EV4F**).

2. Indicate which ASO is targeting the C141Y region in human fibroblasts.

ASO4 targets C141Y region in human fibroblasts and we have now included the information in **Figure 6B**.

3. The protein nomenclature should be different for smaller protein generated in mouse (SEL1L-B presumably lacking 50 aa) and human (Δ exon4, lacking 56 aa) induced by partial or full exon 4 skipping, respectively.

We acknowledge the reviewer's point that the mouse (*SEL1L-B*, lacking 50 amino acids) and human (Δ exon4, lacking 56 amino acids) forms are distinct. Nevertheless, to maintain clarity and facilitate comparison between species, we have kept the same nomenclature for both the mouse *SEL1L-B* and the ASO-induced exon skipping band. We have clarified this point in the text to highlight the differences between mouse and human *SEL1L-B* in the manuscript.

4. Significantly is misspelled as "significatntly" in the 2nd paragraph of the last results section.

We thank the reviewer for pointing this out and have corrected typos throughout the revised manuscript.

Referee #2:

In the manuscript "Functional rescue of a fatal ERAD mutation via alternative splicing", Wang and colleagues report that alternative splicing of SEL1L can ameliorate pathological phenotypes caused by the bi-allelic SEL1L Cys141Tyr mutation. Using several knock-in mouse models, the authors characterize an alternative splicing event resulting in Sel1l-b mRNA and assess the impact of increased levels in the context of a bi-allelic Sel1l C141 mutation on survival, ENDI-like phenotypes, SEL1L/HRD1 expression, ERAD function, and UPR activation. They further evaluate antisense oligonucleotide-mediated exon skipping in patient-derived fibroblasts as a potential therapeutic strategy to counteract SEL1L Cys141Tyr-associated pathology.

The manuscript is well structured, clearly written, and focused. The findings are of broad interest, provide new insight into SEL1L function, and suggest avenues for therapeutic intervention. I have no major concerns and consider the study, in principle, suitable for publication in EMBO J. Below suggestions to further strengthen the work before publication:

We sincerely thank the reviewer for the positive evaluation of our work. We greatly appreciate the constructive suggestions and comments, which have helped us further strengthen the manuscript and will also guide our future studies.

Minor

To increase the value for basic research and to substantiate the finding of functionality of the short SEL1L isoform and potentially the role of the FNII domain, I suggest to add a comparative global proteomic mass-spectrometry dataset from patient-derived fibroblasts (plus minus) ASO1, ideally benchmarked against family-derived wild-type fibroblasts. As an alternative or complement, comparative global proteomes from WT vs. KI vs. KI' liver or brain tissue would be highly informative.

We truly appreciate the reviewer's insightful suggestion. We agree that a comparative global proteomic mass-spectrometry dataset, either from patient-derived fibroblasts or from KI and KI' tissues, would provide valuable information to further substantiate the functionality of the short SEL1L isoform and the potential role of the FNII domain. Indeed, we are performing these experiments as an important direction to investigate the molecular and physiological mechanisms of SEL1L-HRD1 ERAD.

Nonessential

- A mass-spectrometry interactome comparing long vs. short SEL1L would be valuable for the same reasons. However, depending on tool availability (e.g. functionality of the abcam antibody in IP), this may fall outside the scope of the present study.

We appreciate the reviewer's comments. This is indeed an important avenue for future investigation to further clarify the functional role of the short SEL1L isoform.

- Throughout the manuscript, I suggest to avoid presenting IRE1 solely as an ERAD substrate. Because readers less familiar to the system may not be able to grasp the full picture. Better explicitly link IRE1 also to the UPR (e.g., 'the UPR sensor and ERAD substrate, IRE1') and cite an appropriate reference. Reviewer expertise: cell biology with a focus on ER quality control; limited expertise in in-vivo mouse diagnostics and splicing mechanisms.

We thank the reviewer for this helpful suggestion. We agree that it is important to describe IRE1 α both as a UPR sensor and as an ERAD substrate, to provide a complete context for readers less familiar with the system. We clarify this point in the manuscript and included appropriate references.

Referee #3:

The manuscript by Wang H. et al. investigates the pathogenic mechanism of ERAD-associated neurodevelopmental disorders with onset in infancy (ENDI) and explores potential therapeutic strategies. Previous work from this group linked recessive mutations in SEL1L and HRD1 to ENDI. To model the disease, the authors used CRISPR to introduce a C141Y point mutation in SEL1L in mice. Most homozygous mutants died shortly after birth, but one line survived to adulthood without obvious defects. Mechanistic analysis revealed that this line carried an additional splice-donor mutation, producing a truncated SEL1L protein lacking a fibronectin domain yet retaining ERAD activity. This truncated protein rescued the developmental defects otherwise caused by SEL1L loss. The authors further validated these findings in patient-derived fibroblasts carrying the same mutation.

Overall, this study provides strong evidence that the C141Y mutation destabilizes SEL1L, impairing ERAD and leading to developmental defects. The discovery that an alternatively spliced SEL1L variant retains function and rescues the phenotype offers important new insight and suggests a potential therapeutic avenue for this devastating disease. I have only a few minor suggestions for revision.

We sincerely appreciate the reviewer's thoughtful suggestions, which have helped us to further improve the manuscript.

Specific points:

1. The alternative splicing-based mechanism described here appears to be unique to this case, as most truncated proteins are non-functional. In the abstract, the authors should avoid overstating its broader therapeutic potential for other genetic disorders.

We appreciate the reviewer's suggestion. We have now revised the abstract accordingly.

2. Please spell out "ERAD-associated neurodevelopmental disorders with onset in infancy (ENDI)" at its first mention.

We have revised the text accordingly.

3. Page 3: replace "largely uncoupled from ER stress response" with "largely independent of the ER stress response."

We have revised the text accordingly.

4. Next paragraph: "bi-allelic SEL1L and HRD1 variants" should be revised to "bi-allelic SEL1L or HRD1 variants." The same correction applies to the following sentence.

We have revised the text accordingly.

5. Page 4: the phrase "three independent founder lines" is unclear. Does this refer to three parental lines of distinct genetic backgrounds, or to three lines of the same background?

We thank the reviewer for pointing out this ambiguity. To clarify, our mouse line was generated using CRISPR-Cas9, and the three independent founder lines were derived from three separate embryo injection experiments, all on the same genetic background (C57BL6/J). We have revised the Main text and Methods section accordingly to avoid confusion.

6. The sentence "Given their shared phenotypes..." is ambiguous. Since the prior sentence refers to heterozygous mice from all lines, does "their" refer to these heterozygous lines?

We have revised the text accordingly as "Given the shared phenotypes between Line A and B, they were grouped as the KI line for subsequent analyses using tissues from p0 neonates due to lethality."

7. Figure 1H: truncated SEL1L levels appear comparable to full-length SEL1L in wild-type animals, yet the quantification in Figure 1I indicates only partial rescue. Please clarify.

We appreciate the reviewer's comments. We now have repeated the experiment and showed that overall expression of the truncated Sel1L protein in KI' mice was consistently lower than in WT controls, as detected using two different antibodies (**Response Figure 5**). This observation is consistent with our previous findings that FNII-deleted SEL1L is less stable than full-length SEL1L in HEK293T cells (Weis *et al*, 2024).

Response Figure 5. SEL1L expression in WT, KI, and KI' mice. Western blot analysis of SEL1L protein level in P0 mouse livers. Two SEL1L antibodies (Ab1, Ab2) were from Abcam and home-made, respectively.

8. Figure 2C: the intron labels are confusing. Why are they designated as 3' or 5'?

The original labels in the plasmid construct indicated the 5' and 3' ends of the Adeno intron flanking the inserted mini exon. Because these labels did not provide additional information and could lead to confusion, we have removed them. The 3' and 5' labels shown for the *mSel1L* intron refer specifically to the 3' end of intron 3 and the 5' end of intron 4, which are immediately adjacent to Exon 4.

9. Figure 4B: this figure is not mentioned in the main text.

We have now added the corresponding reference in the main text.

10. Page 9: the sentence "although disulfide bonds within the FNII domain are critical for SEL1L function" could be clarified as "although disulfide bonds within the FNII domain are critical for SEL1L stability when the FNII domain is present."

We have revised the text accordingly.

11. Citation issues:

Lilley & Ploegh (2004) does not discuss SEL1L. It was entirely focused on Derlin-1.

Oda et al. (2006) focuses on Derlins, not SEL1L.

For SEL1L's role in HRD1 stability, the authors should cite PMID: 23867461 (see Figure 5).

For SEL1L's role in substrate recruitment, they should cite Daniel Hebert's work (PMID: 19524542). The reference "Wang HH" should be updated, as the work is now published.

We appreciate the reviewer's comments. We have updated the citations accordingly.

References:

Ibrahim EC, Schaal TD, Hertel KJ, Reed R, Maniatis T (2005) Serine/arginine-rich protein-dependent suppression of exon skipping by exonic splicing enhancers. *Proceedings of the National Academy of Sciences of the United States of America* 102: 5002-5007

Long JC, Caceres JF (2009) The SR protein family of splicing factors: master regulators of gene expression. *Biochem J* 417: 15-27

Shepard PJ, Hertel KJ (2009) The SR protein family. *Genome Biol* 10: 242

Tacke R, Manley JL (1995) The human splicing factors ASF/SF2 and SC35 possess distinct, functionally significant RNA binding specificities. *EMBO J* 14: 3540-3551

Weis D, Lin LL, Wang HH, Li ZJ, Kusikova K, Ciznar P, Wolf HM, Leiss-Piller A, Wang Z, Wei X *et al* (2024) Biallelic Cys141Tyr variant of SEL1L is associated with neurodevelopmental disorders, agammaglobulinemia, and premature death. *J Clin Invest* 134: e170882

Dear Prof. Qi,

Thank you for submitting a revised version of your manuscript. Your study has now been seen by one of the original referees, who finds that their previous concerns have been addressed and now recommends publication of the manuscript. There remain only a few mainly editorial points that have to be addressed before I can extend formal acceptance of the manuscript:

- Please remove the figures from the manuscript file and upload as individual Figure. Please also remove the track changes;
 - Please move the "MATERIALS AND METHODS" section from the Appendix into the manuscript file
 - Please select at least one contribution for Emily Whitestone in eJP
 - Please add an ORCID ID for Shengyi Sun and Huilun Wang
 - Please add the FUNDING INFO for HHS | NIH | National Institute of General Medical Sciences (NIGMS) R35GM130292 into the manuscript
 - Please reduce the number of keywords to 5) and place below Abstract
 - Please rename the Conflict of Interest section into "Disclosure and Competing Interests Statement", in accordance with our updated Guide to Authors (<https://link.springer.com/partners/embo-press/editorial-policies#Competing%20interest%20disclosures>)
 - As we are switching from a free-text author contribution statement towards a more formal statement based on Contributor Role Taxonomy (CRediT) terms, please remove the present Author Contribution section and instead specify each author's contribution(s) directly in the Author Information page of our submission system during upload of the final manuscript. See <https://casrai.org/credit/> for more information
- There is a reference to "data not shown" on page 4, "Sequence analysis confirmed that no off-target mutations were present in the Sel1L cDNA aside from the engineered mutations (data not shown)". According to our policy, which does not permit references to "data not shown", please include this information in the Appendix. Please see also <https://www.embopress.org/page/journal/14602075/authorguide#unpublisheddata>.
- Please adjust the in-text callouts for individual figures and figure panels: e.g. Supplementary Table 1 (that should be renamed to Appendix Table S1) seems to be missing
 - APPENDIX 1 FILE WITH ToC: Appendix file needs to be in PDF format; title page should contain "Appendix for + ms title" and ToC with the page numbers for the listed items; nomenclature should be Appendix Figure Sx and Appendix Table Sx throughout ms and Appendix PDF
 - Please remove the R&T TABLE from Appendix, and only leave it uploaded individually
 - Please zip the SD for Fig. 5H in 3 separate zip folders due to the upload size up to 300MB, labeled as Fig. 5H-1, 5H-2 and 5H-3)
 - Please provide suggestions for a short 'blurb' text prefacing and summing up the conceptual aspect of the study in two sentences (max. 250 characters), followed by 3-5 one-sentence 'bullet points' with brief factual statements of key results of the paper; they will form the basis of an editor-written 'Synopsis' accompanying the online version of the article. Please also provide an altered synopsis image, making sure that the aspect ratio conforms to our website's format - it should be exactly 550 pixels wide and between 300-600 pixels high.
 - Figure Legends (main + EV): Please note that the exact p values are not provided in the legends of figures 1F, J; 2D, E; 3G, J; 4A, E, F, H, J; 5C, E, I; 6E, 7A, E, G, F; EV2 A, D, E; EV4 A.
 - Sections need to be named and the order should be corrected: Title page - Abstract - Keywords - Introduction - Results - Discussion - Methods - Data Availability - Acknowledgements - Disclosure and Competing Interests Statement - References - Figure Legends - Table(s) - Expanded View Figure Legends.

With best regards,
Cornelius Schneider

Cornelius Schneider, PhD
Editor | The EMBO Journal
c.schneider@embojournal.org

Please refer to our figure preparation guideline in order to ensure proper formatting and readability in print as well as on screen:

<https://link.springer.com/journal/44318/submission-guidelines#cms-Figure-and-data-presentation>

Referee #1:

The authors have addressed all of my concerns. Nice work!

All minor editorial requests have been addressed by the authors.

Dear Prof. Qi,

I am pleased to inform you that your manuscript has been accepted for publication in the EMBO Journal.

You may qualify for financial assistance for your publication charges - either via a Springer Nature fully open access agreement or an EMBO initiative. Check your eligibility: <https://link.springer.com/journal/44318/how-to-publish-with-us>

Yours sincerely,

Cornelius Schneider, PhD
Editor
The EMBO Journal
c.schneider@embojournal.org

Please note that it is The EMBO Journal policy for the transcript of the editorial process (containing referee reports and your response letters) to be published as an online supplement to each paper. If you should prefer removal of any referee-only figures included in the point-by-point response(s), e.g. because they may still be used for future publication or because they have been reproduced from published work by others, please do let us know immediately via response email.

More information is available here: <https://link.springer.com/partners/embo-press/editorial-policies#Peer%20review>